# Evidence from Finland and Sweden on the relationship between early-life diseases and lifetime childlessness in men and women

Aoxing Liu [1,2,3] ✉, Evelina T. Akimova [4], Xuejie Ding[4], Sakari Jukarainen [1], Pekka Vartiainen [1], Tuomo Kiiskinen [1,5], Sara Koskelainen[1], Aki S. Havulinna [1,5], Mika Gissler [5,6,7,8], Stefano Lombardi [9], Tove Fall [10], Melinda C. Mills [4,11,12] ✉ & Andrea Ganna [1,2,3] ✉

The percentage of people without children over their lifetime is approximately 25% in men and 20% in women. Individual diseases have been linked to childlessness, mostly in women, yet we lack a comprehensive picture of the effect of early-life diseases on lifetime childlessness. We examined all individuals born in 1956–1968 (men) and 1956–1973 (women) in Finland (*n* = 1,035,928) and Sweden (*n* = 1,509,092) to the completion of their reproductive lifespan in 2018. Leveraging nationwide registers, we associated sociodemographic and reproductive information with 414 diseases across 16 categories, using a population and matched-pair case–control design of siblings discordant for childlessness (71,524 full sisters and 77,622 full brothers). The strongest associations were mental–behavioural disorders (particularly among men), congenital anomalies and endocrine–nutritional–metabolic disorders (strongest among women). We identified new associations for inflammatory and autoimmune diseases. Associations were dependent on age at onset and mediated by singlehood and education. This evidence can be used to understand how disease contributes to involuntary childlessness.

In western European countries, about 20% of women born around 1965 remained childless by 2010; the highest levels of lifetime childlessness are now in East Asia, ranging from 28% (Japan) to 35% (Hong Kong) for women born in 1975[1,2]. There has also been a sharp increase in childlessness over the past few decades in countries such as Finland, increasing since the 1970s from 14% to 22% (women aged 40 years) and 22% to 32% (men)[3,4]. Demographic research has isolated core factors linked to the rise in contemporary levels of childlessness. Access to effective contraception at the end of the 1960s in many countries and, more recently, emergency post-coital contraception

[1]Institute for Molecular Medicine Finland, University of Helsinki, Helsinki, Finland. [2]Program in Medical and Population Genetics, Broad Institute of MIT and Harvard, Cambridge, MA, USA. [3]Analytic and Translational Genetics Unit, Massachusetts General Hospital, Boston, MA, USA. [4]Leverhulme Centre for Demographic Science, Nuffield Department of Population Health, and Nuffield College, University of Oxford, Oxford, UK. [5]Finnish Institute for Health and Welfare, Helsinki, Finland. [6]Research Centre for Child Psychiatry and Invest Research Flagship, University of Turku, Turku, Finland. [7]Academic Primary Health Care Centre, Stockholm, Sweden. [8]Department of Molecular Medicine and Surgery, Karolinska Institutet, Stockholm, Sweden. [9]VATT Institute for Economic Research, Helsinki, Finland. [10]Molecular Epidemiology, Department of Medical Sciences, and Science for Life Laboratory, Uppsala University, Uppsala, Sweden. [11]Department of Economics, Econometrics and Finance, Faculty of Economics and Business, University of Groningen, Groningen, the Netherlands. [12]Department of Genetics, University Medical Centre Groningen, Groningen, the Netherlands. ✉e-mail: aoxing.liu@helsinki.fi; melinda.mills@demography.ox.ac.uk; andrea.ganna@helsinki.fi

in the late 1990s provided couples with greater ability to control their reproduction[1]. Women's opportunities also vastly changed in many industrialized nations since the 1980s, with gains in higher levels of education and entry into the labour market accompanied by shifts in gender norms and equity[5,6]. At the same time, both men and women faced work–life reconciliation constraints when planning to have children, including lack of access to childcare, challenging housing conditions, economic uncertainty, inability to find a partner and the general absence of supportive family policies[6]. Given that women entered the labour market while still taking on the bulk of household labour, this 'incomplete gender revolution' resulted in women often being forced to decide between a career and parenthood[7,8]. Higher levels of job strain and lack of work–life reconciliation have also been found to reduce fertility intentions[9]. Another shift, which has been particularly linked to men, is the growth of precarious, temporary and uncertain employment, leading many to postpone or forgo entering partnerships and parenthood[10,11]. Many of these changes resulted in a general postponement of children to later ages[6]. This in turn meant that individuals were having children at ages when they had lower fecundity (reproductive capacity), leading to infertility-related issues and having fewer children[12]. This also resulted in an increase in involuntary childlessness (Box 1). The rise in childlessness among men includes many of the factors above, such as the role of economic uncertainty, socio-economic circumstances and structural challenges such as childcare. More recently, key predictors for male childlessness are the rise of multiple and short partnerships[4] and a normative shift to a growing group of men who are uninterested in becoming fathers[13]. Others have linked rises in male childlessness to lifestyle factors, such as becoming more sedentary, alcoholism, tobacco use and poor diet[14].

Although economic, societal and cultural freedom has been linked to individuals remaining childfree or voluntarily childless[15], the percentage of individuals who report that they never intended to have children has remained low and is estimated to be around 5% in Europe[16]. Rather than being planned at an early age, becoming voluntarily childfree may be a mix of unforeseen circumstances, often related to the postponement of childbearing and adaptation to a childfree life[17,18]. This makes the decomposition of measuring voluntary versus involuntary childlessness challenging. A longitudinal study[19] measuring the fertility intentions of childless women (questioned from age 14 to their late 40s) found that they engaged in a repeated postponement of childbearing and the subsequent adoption of a childless expectation at older ages or had indecision about parenthood evidenced by changing childless expectations across various ages.

The majority of childless individuals appear to be involuntarily childless and wanted to have children. From the standpoint of public health, involuntary childlessness may impact other health domains. Childless men and women are more likely to suffer from relationship dissolution, lower levels of self-esteem and isolation, and higher risks of clinical depression[6,20]. A recent systematic review found that infertile women, particularly those in middle- and low-income countries, are more likely to experience psychological, physical and sexual violence as well as economic coercion[21]. For childless men, their workplace interactions and career opportunities have been found to be negatively affected, whereas psychological wellness is rarely addressed[20].

While the aforementioned demographic and socio-economic literature has shown that social, economic and structural factors have significant effects on childlessness, the fact that a large proportion of individuals experience this involuntarily suggests that diseases might play a role in influencing individuals' chances of being childless or having a particular number of children over their lifetime (Supplementary Fig. 1). First, diseases may directly influence childlessness through medical conditions that affect the fecundity, co-morbidities and mortality of the individual, as well as the risk of stillbirth[22]. Second, diseases can impact selection into a partnership, which in turn lowers the chance of being partnered and thus having children[23]. Third, some may choose

## BOX 1

# Defining childlessness

We use the term 'childlessness' to describe the state of individuals that have had no live-born children by the end of their reproductive lifespan (age 45 for women[32,62,63] and age 50 for men[32,63]). Childlessness is defined in the literature as being voluntary or childfree[43] (for example, active choice or preference[70]), involuntary (for example, infertility, stillbirth or reproductive-age mortality) or a mix of the two, such as circumstantial situations related to partnership and socio-economic status (Supplementary Fig. 1). Although researchers have conducted in-depth studies showing the measurement challenges of differentiating the two, it has been estimated that, in Europe, around 4–5% of individuals within reproductive ages do not intend to have children[16]. This together with the overall lifetime childlessness proportion being 15–20%[1] suggests that around three quarters of childless individuals remain so involuntarily. The approach of this paper is to provide a data-driven, factual examination of the associations between early-life diseases and lifetime childlessness, with the aim to understand how early disease relates to childlessness among people who want to have children, with the future potential to intervene.

to be childfree due to knowledge or concerns of intergenerational transmission of certain known genetic conditions[24]. Finally, many diseases are associated with lower socio-economic status, unemployment and economic uncertainty, which may amplify the psychosocial effect of diseases on childlessness, possibly by increasing inability to find a partner, increasing partnership instability or reducing fertility intentions[25].

A review of the literature linking disease to childlessness can be found in Supplementary Table 1. Previous medical studies largely examined only a limited selection of diseases directly associated with reproductive biology such as recurrent miscarriage[26], polycystic ovary syndrome[27] and endometriosis[28]. Another large area of research has been the study of infertility and inflammatory bowel disease (IBD)[29,30]. A systematic review found that up to one-third of individuals with IBD opted to be voluntarily childfree, often related to a lack of knowledge related to pregnancy-related IBD issues[29], with clinical studies finding that women with moderate to severe IBD had higher incidences of adverse pregnancy outcomes[30]. Multiple sclerosis has also been associated with higher levels of childlessness, linked to fewer people with multiple sclerosis being in a stable relationship, fear of genetically transmitting the disease or discontinuation of treatments[31]. Another line of study has examined fecundity in patients with major psychiatric disorders[32,33] and in those with a genetic liability for schizophrenia[34]. Overall, for men, comparatively limited information—often from highly selective semen samples from infertility clinics—has been collected on the causes and consequences of infertility and having children[35,36], despite infertility being a major public health concern.

Current research has examined singular diseases in parallel and thus lacks a systematic assessment of the role of multiple diseases on childlessness and their relative importance. Moreover, the association of diseases can be sex-specific, given that men and women differ regarding both reproductive patterns (men have a higher chance of being childless, higher parity and a longer reproductive lifespan than women[37]) and the prevalence and severity of diseases. Finally, previous studies have had a limited time horizon to follow participants and limited methodological tools to control for unmeasured familial confounding, either genetic or environmental.

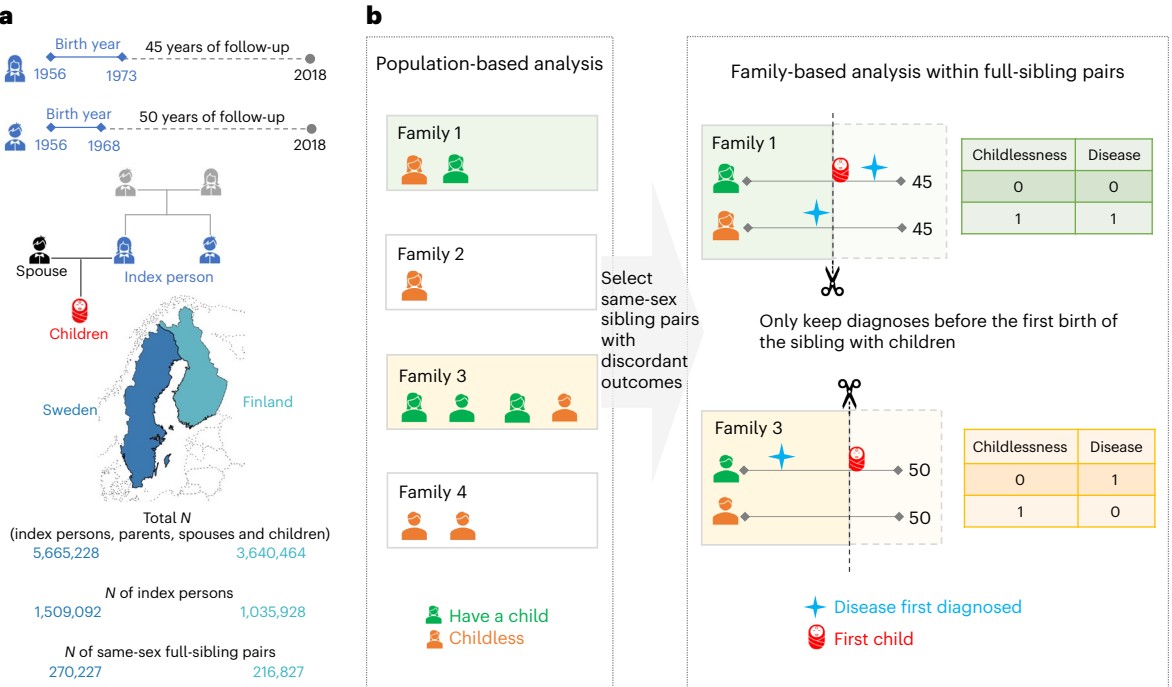

**Fig. 1 | Schematic overview of the study. a**, The birth cohorts, the follow-up period and the number of Finns and Swedes included in the analyses. A total of three generations of familial relationships were considered: for the index persons, the parental information used to match sibling pairs, the children's information to define the main outcome of childlessness and the registered spouse's information to determine the secondary outcome of singlehood. Map adapted with permission from MapChart. **b**, The statistical approach used in the main analyses, where only families with same-sex siblings discordant for childlessness were included. Within each family, we randomly selected one childless sibling as a case, and, as the control, the sibling with children that was the closest in birth order to the case. Within a sibling pair, disease diagnoses were considered only if they occurred at least one year before the birth of the first child or within the corresponding age at which the sibling was childless, since sibling ages differ.

We address these limitations of previous research by studying the entire reproductive and health history of a cohort of Finnish and Swedish men and women until the end of their reproductive period using high-quality nationwide population registers. The current study determines to what extent diseases are associated with childlessness; whether such associations vary between sexes, differ by the age at disease onset or stratify by parity; and the roles that partnership formation and educational level play in mediating the results. To answer these questions, we used a disease-agnostic family-based approach, which allows us to robustly evaluate the relative importance of 414 disease diagnoses in relation to childlessness while providing public-health-relevant metrics to interpret the relative importance and contextualize the results of our study.

## Results

### Study population

Our study population included 1,425,640 women and 1,119,380 men, who were born after 1956 and alive by age 16, did not emigrate, and largely completed their reproductive period by the end of 2018 (Fig. 1a). Among them, 230,198 women and 279,454 men remained childless. The proportion of childless individuals—those with no live-born children—was higher in men (25.4%) than women (16.6%), higher among Finns (23.3%) than Swedes (18.7%) and slightly higher in the general population than in individuals who had same-sex full siblings (0.9% higher in women and 0.5% higher in men). Among individuals with children, a two-child parity was most prevalent in men (47.1%) and women (48.9%), among both Finns (44.0%) and Swedes (50.8%) (Supplementary Table 2). Both women and men with the lowest educational attainment were more likely to be childless (for example, in Finland, 24.2% were childless in women and 37.4% in men) than the general population (for example, in Finland, 18.8% were childless in women and 27.7% in men) (Supplementary Tables 3 and 4).

When looking across generations, the education level of an individual's parents was correlated with that individual's childless status (Supplementary Tables 3 and 4). For example, among index persons whose parents had completed the first stage of tertiary education, men were slightly less likely to be childless (for example, in Finland, 25.3%) than the general population (for example, in Finland, 27.7%), while women behaved in the opposite way (for example, in Finland, 21.3%, compared to 18.8% in the general population).

For the entire study population, we have information from nationwide registers covering 414 disease diagnoses across 16 main categories (Supplementary Fig. 2 and Supplementary Table 5). Disease prevalence across the entire follow-up was highly comparable between Finland and Sweden (Supplementary Fig. 3), and we present meta-analysis results between the two countries. Our main analysis focused on 71,524 full-sister and 77,622 full-brother pairs who were discordant on childlessness (Fig. 1b).

### Disease diagnoses associated with childlessness

Eleven rare diseases were associated with an almost complete lack of children (for example, severe intellectual disability, childhood leukaemia and muscular dystrophy) (Supplementary Table 6) and were therefore not included in the remaining analyses. Of the remaining 403 diseases (328 in women and 325 in men), 74 were significantly associated with childlessness in at least one sex ($P < 1.5 \times 10^{-4}$, after multiple-testing correction), including 33 disease diagnoses shared among women and men (Fig. 2). A full list of the results can be found in Supplementary Table 7 or in an interactive dashboard: https://dsgelrs.shinyapps.io/DiseaseSpecificLRS/. More than half of the significant associations were mental–behavioural disorders (26/53 (49%) in women and 30/54 (55.5%) in men), which was, together with congenital anomalies, the disease categories with the strongest associations with childlessness

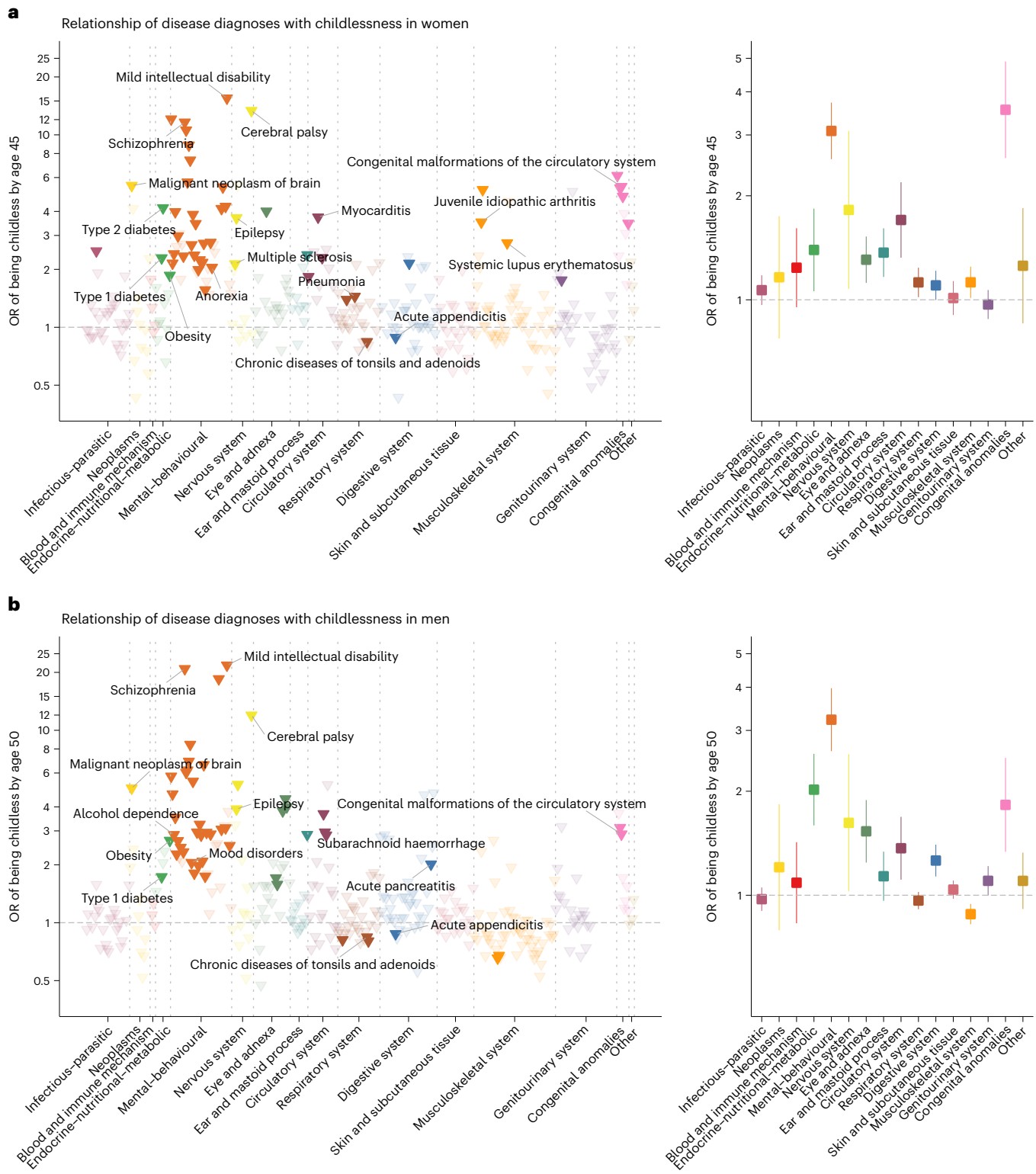

**Fig. 2 | Relationship of 403 disease diagnoses with childlessness by age 45 in women and age 50 in men in 71,524 full-sister and 77,622 full-brother pairs who were discordant on childlessness, using a matched case–control design. a,b,** Odds ratios (ORs) are computed for each disease diagnosis (left) and averaged over disease categories (right). The error bars indicate 95% CIs. Only disease diagnoses that are significantly associated with childlessness after multiple-testing correction ($P < 1.5 \times 10^{-4}$) are coloured. Labels are assigned only for certain disease diagnoses that are described in the text.

(in women: odds ratio (OR) = 3.1; 95% confidence interval (CI) (2.6–3.7); in men: OR = 3.2; 95% CI (2.6–4.0)), averaging over all mental–behavioural disorders (Fig. 2 and Supplementary Table 8). There was substantial heterogeneity among different mental–behavioural disorders

($P < 0.001$ in both women and men in a heterogeneity test). For example, mild intellectual disability was the condition with the strongest association with childlessness (for example, in men, OR = 21.7; 95% CI (10.8–43.5)), while smaller effects were observed for mood disorders

(for example, in men, OR = 2.1; 95% CI (1.8–2.3)). Severe diseases of the brain such as cerebral palsy (in women: OR = 13.4; 95% CI (7.9–22.7); in men: OR = 12.0; 95% CI (6.5–22.2)) and malignant neoplasm of the brain (in women: OR = 5.5; 95% CI (2.8–10.8); in men: OR = 5.0; 95% CI (2.8–8.9)), which impair health and functioning in several ways but can also result in behavioural and personality disorders[38,39], were also strongly associated with childlessness. For individuals with severe mental disorders or physical disabilities (for example, intellectual disability), especially among women, involuntary sterilization was historically carried out in Finland (from 1935 to 1970) and Sweden (1934 to 1976)[40].

Another disease category strongly associated with childlessness, in both sexes, encompassed the endocrine–nutritional–metabolic disorders (in women: OR = 1.4; 95% CI (1.1–1.8); in men: OR = 2.0; 95% CI (1.6–2.6); averaging over all endocrine–nutritional–metabolic disorders). For example, obesity (in women: OR = 1.8; 95% CI (1.5–2.4); in men: OR = 2.7; 95% CI (2.0–3.6)), which is normally recorded in the secondary health-care registers only in severe cases, and type 1 and type 2 diabetes (for example, in women, OR = 2.3; 95% CI (2.0–2.6) for type 1 diabetes and OR = 4.2; 95% CI (2.2–8.1) for type 2 diabetes) were among the diseases with the strongest associations with childlessness.

We found several inflammatory diseases to be significantly associated with childlessness across multiple organ systems including the respiratory, circulatory, genitourinary, digestive, nervous and musculoskeletal systems. For example, a diagnosis of pneumonia (OR = 1.4; 95% CI (1.3–1.6)), myocarditis (OR = 3.7; 95% CI (2.3–6.1)), chronic tubulo-interstitial nephritis (OR = 1.8; 95% CI (1.3–2.3)), multiple sclerosis (OR = 2.1; 95% CI (1.5–3.1)), systemic lupus erythematosus (OR = 2.7; 95% CI (1.6–4.6)) or juvenile idiopathic arthritis (OR = 3.5; 95% CI (2.6–4.9)) significantly increased subsequent childlessness in women. Puzzlingly, but consistent with previous findings[41], we observed that chronic diseases of the tonsils and adenoids and acute appendicitis were associated with reduced rather than increased odds of childlessness for women and men (for example, for chronic diseases of the tonsils and adenoids, in women: OR = 0.84; 95% CI (0.80–0.88); in men: OR = 0.84; 95% CI (0.80–0.89)), with similar effect sizes in Finns and Swedes (P for heterogeneity, 0.60 between the two countries in both sexes).

## Sex-specific effects

The disease category encompassing congenital anomalies showed markedly different effects between sexes (in women: OR = 3.5; 95% CI (2.6–4.9); in men: OR = 1.8; 95% CI (1.3–2.5); P for sex difference, 3.7 × 10$^{-3}$). Overall, malformations of the digestive system (in women: OR = 4.8; 95% CI (2.2–10.3); in men: OR = 1.7; 95% CI (1.0–3.0); P = 0.03) and the musculoskeletal system (in women: OR = 3.5; 95% CI (2.0–6.1); in men: OR = 1.2; 95% CI (0.7–1.9); P = 3.7 × 10$^{-3}$) were more strongly associated with childlessness in women than in men.

Significant sex-dependent effects were also observed for several mental–behavioural and endocrine–nutritional–metabolic disorders (Fig. 3a). For example, diagnoses of schizophrenia (in women: OR = 11.6; 95% CI (9.1–14.9); in men: OR = 20.8; 95% CI (16.1–27.0); P for sex difference, 1.5 × 10$^{-3}$) and acute alcohol intoxication (in women: OR = 2.2; 95% CI (1.6–2.9); in men: OR = 4.7; 95% CI (3.8–5.7); P < 0.001) had stronger associations with childlessness in men than in women. Diagnosis of type 1 diabetes showed a stronger association in women than in men (in women: OR = 2.3; 95% CI (2.0–2.6); in men: OR = 1.7; 95% CI (1.5–2.0); P = 3.9 × 10$^{-3}$). The stronger association in women might be due to the fact that, in the observational period, women with type 1 diabetes were recommended not to get pregnant if under poor glycaemic control[42].

## Age-of-onset of disease effects

We hypothesized that the age when the disease was first diagnosed, a proxy for the age of onset, would influence the chances of being childless by either capturing disease severity or directly impacting factors underlying individuals' reproductive trajectory. We first evaluated whether there was a general trend across disease diagnoses (30 in

women and 31 in men) that were associated with childlessness and for which we had enough individuals to estimate age-of-onset-stratified effects (Fig. 3b and Supplementary Tables 9 and 10). We observed a nonlinear effect, with the strongest association with childlessness occurring when the disease was first diagnosed between 21 and 25 years old in women (OR = 3.1; 95% CI (2.5–3.8)) and later, between 26 and 30 years old, in men (OR = 3.1; 95% CI (2.4–3.9)). Smaller effects were observed at younger ages (for example, in women, OR = 1.9; 95% CI (1.6–2.3) for onset before 16 years old; P < 0.001) and, especially, older ages of onset (for example, in women, OR = 1.2; 95% CI (0.5–2.8) for onset between 36 and 40 years old; P = 0.04). Despite this broader trend, there was substantial heterogeneity between disease diagnoses (Fig. 3c). For example, among women diagnosed with obesity, the group that received their first diagnoses between 16 and 20 years old had higher levels of childlessness than those diagnosed at later ages (OR = 3.0; 95% CI (1.6–5.6) for those diagnosed between 16 and 20 years old; OR = 1.1; 95% CI (0.5–2.7) between 26 and 30 years old; P for differences, 0.08).

## Stratification by parity for individuals with children

In the main analyses, we compared childless individuals to their siblings with children, regardless of parity. We reasoned that childless siblings might be more phenotypically similar to their siblings with fewer children (lower parity) than to siblings who had several children (higher parity). When comparing childless individuals with their siblings with just one child (Supplementary Table 11), 14 disease diagnoses in women and 6 in men had significantly lower odds ratios (Supplementary Fig. 4a,b) than those in the main analysis. For example, in women, the OR of schizophrenia on childlessness dropped from 11.6 (95% CI (9.1–14.9)) to 5.1 (95% CI (3.7–7.0); P < 0.001) when we compared childless individuals with their siblings with just one child. For individuals with higher parities, such as those with exactly two children (Supplementary Fig. 4c,d and Supplementary Table 7) or those with three or more children (Supplementary Fig. 4e,f and Supplementary Table 7), limited differences in ORs were observed compared with the main analyses. Jointly, these results indicate that childless individuals are more similar to their siblings with one child with regard to risk for childlessness-associated early-life diseases than to siblings with higher parities.

## Mediation effect by singlehood

Overall, 83.0% of women and 77.0% of men had registered partners before age 45 and 50, respectively. When restricted to childless individuals, the proportion dropped to 36.3% in women and 29.4% in men. We therefore examined to what degree the associations between disease diagnoses and childlessness were mediated by singlehood. First, we estimated the effect of disease diagnoses on the chance of being without a partner by age 45 in women and age 50 in men (Supplementary Figs. 5 and 6). We then compared log-transformed odds ratios on singlehood with those on childlessness and observed a high correlation between these two estimates in both women ($R^2$ = 0.71, P < 0.001) and men ($R^2$ = 0.85, P < 0.001) (Supplementary Fig. 7). A causal mediation analysis performed for each disease also indicated the mediation role of partnership formation, with 29.3% (median) of the disease effects on childlessness in women and 37.9% in men mediated by singlehood (Supplementary Table 12). Different patterns were observed across diseases. For example, singlehood was a significant mediator for women diagnosed with schizophrenia (OR = 2.1; 95% CI (2.0–2.2) for indirect effect; OR = 6.8; 95% CI (5.0–9.2) for direct effect) but not for women with hypertension (OR = 1.2; 95% CI (1.1–1.4) for indirect effect; OR = 2.4; 95% CI (1.6–3.5) for direct effect).

We next investigated whether some of the associations between disease diagnoses and childlessness remained significant among partnered individuals. We considered 133 disease diagnoses in women and 123 in men, which had more than 30 affected individuals with registered partners in the sibling-based analysis. Some diseases that were strongly associated with singlehood (for example, mild intellectual disability)

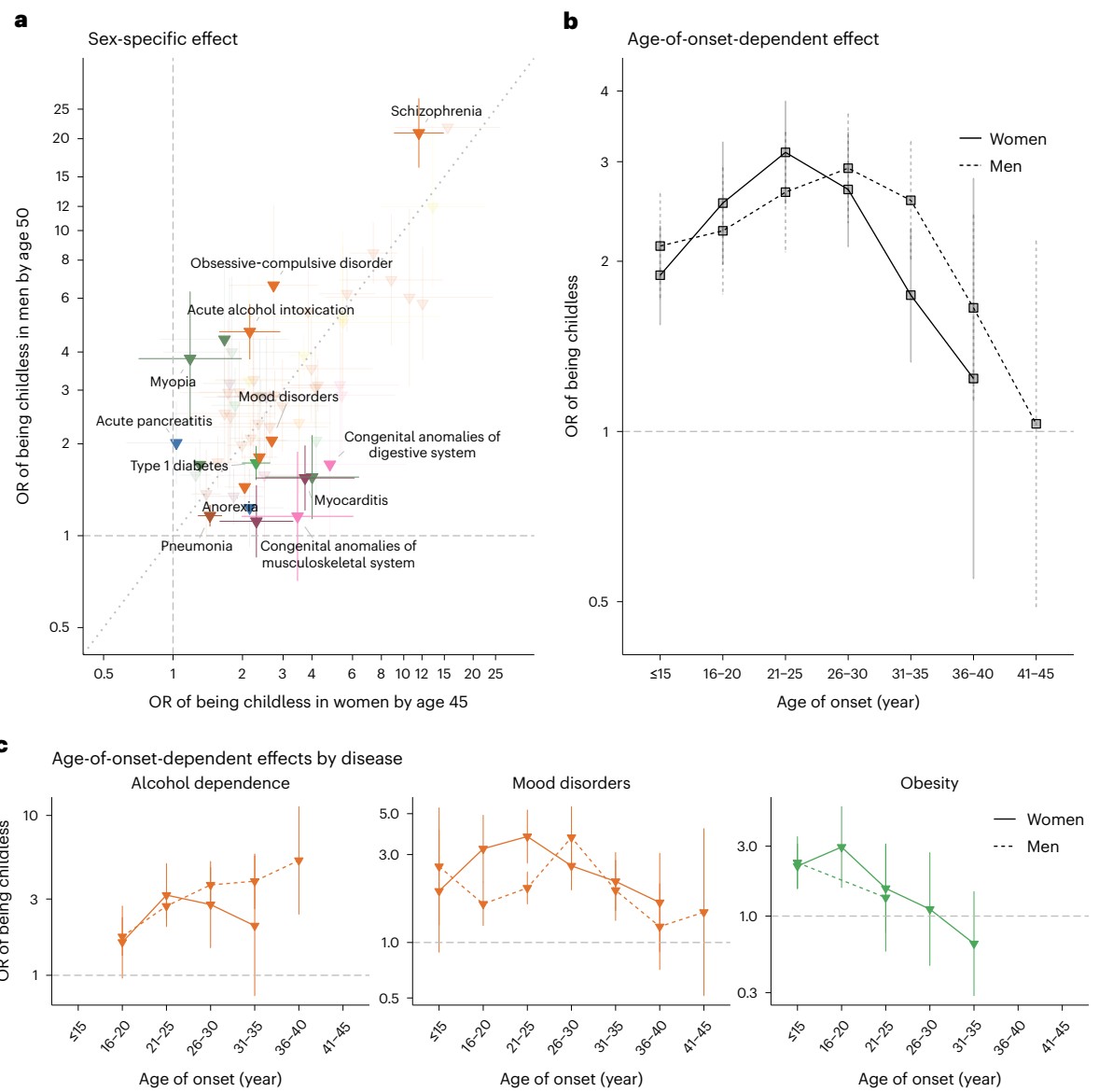

**Fig. 3 | Sex-specific and age-of-onset-dependent effects for the association between disease diagnoses and childlessness in 71,524 full-sister and 77,622 full-brother pairs who were discordant on childlessness. a**, Odds ratios for 60 disease diagnoses that significantly increased the odds of childlessness in either men or women. The error bars indicate 95% CIs. Only disease diagnoses that are significantly different between sexes at a nominal *P* value are coloured. **b**, The average effect associated with childlessness across 30 disease diagnoses in women and 31 in men, for each age category, as obtained from an age-of-onset-stratified analysis. **c**, Age-of-onset-stratified odds ratios associated with childlessness for three major diseases/disorders for which we observe a significant trend. The estimate for each age category is computed only if the number of individuals with this disease within the age group is more than five.

could not be included in the analyses because they were too rare among partnered individuals. Nonetheless, we observed 6 diseases in women and 11 in men that remained associated with childlessness among partnered individuals (Supplementary Fig. 8). Compared with the estimates obtained from both partnered and unpartnered individuals, the effects on childlessness were largely reduced for most disease diagnoses (Supplementary Fig. 9), such as epilepsy (for example, from OR = 3.7; 95% CI (3.2–4.3) for all women to OR = 1.8; 95% CI (1.4–2.4) for partnered women) and acute alcohol intoxication in men (from OR = 4.7; 95% CI (3.8–5.7) for all men to OR = 2.4; 95% CI (1.5–3.8) for partnered men). Several endocrine–nutritional–metabolic diseases retained a strong association with childlessness among partnered individuals, such as obesity in men (all men: OR = 2.7; 95% CI (2.0–3.6); partnered men: OR = 3.3; 95% CI (1.9–5.7)) and type 1 diabetes in women (all women: OR = 2.3; 95% CI (2.0–2.6); partnered women: OR = 2.5; 95% CI (2.0–3.2)).

## Population-based and sensitivity analyses

We performed several sensitivity analyses to determine the robustness of our estimation. First, our main results were based on within-sibling analysis, which can account for unmeasured familial factors but also made several assumptions, including siblings being generalizable to the population. We thus also obtained population-based estimates from a matched case–control design controlling for several potential confounders (Supplementary Fig. 10). Overall, the population-based design resulted in larger sample sizes, similar ORs and smaller CIs compared with the sibling-based design (Supplementary Fig. 11). However, stronger associations were observed from the population-based design in men for psychoactive substance abuse (for example, for alcohol dependence, OR = 2.9; 95% CI (2.5–3.2) from the sibling-based design and OR = 3.8; 95% CI (3.5–4.0) from the population-based design; *P* < 0.001) and mood disorders (OR = 2.1; 95% CI (1.8–2.3)

from the sibling-based design and OR = 2.7; 95% CI (2.5–2.9) from the population-based design; $P$ < 0.001).

Second, in the main analysis, we included individuals who died before the end of follow-up (affecting 7.4% of full sisters and 15.0% of full brothers), thus including reproductive-age mortality as one of the possible mechanisms explaining childlessness. To understand the overall effect of reproductive-age mortality, we also conducted an analysis only considering individuals alive by the end of follow-up (Supplementary Fig. 12). Overall, the results were consistent (Supplementary Fig. 13), but we observed two disease diagnoses in men for which the main effects on childlessness were partially explained by reproductive-age mortality: acute alcohol intoxication (all men: OR = 4.7; 95% CI (3.8–5.8); men alive by age 50: OR = 3.9; 95% CI (2.2–3.8); $P$ = 0.006) and subarachnoid haemorrhage (all men: OR = 2.9; 95% CI (1.8–4.5); men alive by age 50: OR = 0.9; 95% CI (0.4–1.8); $P$ = 0.005).

Third, in the main analysis, we opted for conditional logistic regression because the length of follow-up would be identical for the sibling pairs if ignoring reproductive-age mortality (Figs. 1 and 2). When using a Cox proportional hazards model that captured both time-varying effects and death (Supplementary Fig. 14), we observed similar results with reductions in some disease diagnoses (Supplementary Fig. 15).

Fourth, research has found that individuals who remain childless in Finland, particularly men, have lower levels of education[43,44], suggesting that it may be important to adjust for educational level. In the sibling-based analysis, we adjusted for differences in individuals' education levels between siblings but did not find substantial differences in our results compared with the unadjusted main analysis (Supplementary Fig. 16). In the population-based analysis, as expected, significant changes in ORs were observed for many diseases, in both women and men, when we adjusted for individuals' highest education level (Supplementary Fig. 17). For example, in men, after adjusting for the highest education level, many disease diagnoses had significantly reduced ORs (16 of 45 (35.5%) disease diagnoses with reduced odds of childlessness), while obesity (OR increased from 3.2 (95% CI (2.8–3.7)) to 4.4 (95% CI (3.7–5.2)); $P$ differences, $3.5 \times 10^{-3}$) and acute pancreatitis (OR increased from 1.9 (95% CI (1.6–2.3)) to 2.5 (95% CI (2.1–3.0)); $P$ differences, 0.04) were the only disease diagnoses exhibiting stronger associations.

## Discussion

The rich longitudinal Finnish and Swedish nationwide population registers provided a unique resource to comprehensively assess the associations of several diseases with childlessness and the relative strength of these associations. In addition to the results summarized here, all findings can be explored on the interactive online dashboard: https://dsgelrs.shinyapps.io/DiseaseSpecificLRS/.

Our study used a hypothesis-free approach to estimate the associations between early-life diseases and childlessness across the entire reproductive lifespan. Importantly, we identified several new disease–childlessness associations such as autoimmune diseases (for example, juvenile idiopathic arthritis, multiple sclerosis and systemic lupus erythematosus) and inflammatory diseases (such as myocarditis). For certain diseases, treatments such as methotrexate (an immunosuppressant widely used for rheumatoid arthritis and multiple sclerosis, and occasionally used for systemic lupus erythematosus) have been reported to have side effects on fertility in women[45].

Previous literature has given much attention to the role that diseases play in childlessness among women, whereas men have been understudied[46]. We observed substantial sex differences in disease–childlessness associations. For example, mental–behavioural disorders such as schizophrenia and acute alcohol intoxication had stronger associations with childlessness in men than in women, whereas diabetes-related diseases and congenital anomalies had stronger associations among women. These sex-specific effects may be explained by differences in disease severity, bias in diagnostic practices, partner

history and direct biological effects on fertility. Sex differences were also observed in the association of the age of onset of disease with childlessness. Overall, diseases diagnosed between 21 and 25 years old in women and between 26 and 30 years old in men tend to have the strongest association with childlessness. This is also consistent with the sex difference in age at union formation, which occurs later in men[47].

Our analyses highlight the important associations between many diseases and partnership formation. Childless individuals are twice as likely to be single, in line with previous research showing the importance of selection into partnership on fertility[4,23]. Singlehood represents a major mediator for the odds of being childless, especially in men, and we estimated that 29.3% (median) of the disease effect on childlessness in women and 37.9% in men was mediated by partnership formation. The link between disease diagnoses and partnership formation is probably mediated by complex sociocultural factors, social norms and sex-specific behavioural preferences[47–49]. For example, in the current study, we found that mental–behavioural disorders such as schizophrenia and acute alcohol intoxication are strongly associated with singlehood in men. The effects are substantially smaller in women, which probably reflects how different social and individual preferences for behavioural phenotypes correlated with mental–behavioural disorders impact partner choice in the two sexes[48].

Nonetheless, we also identified several diseases that are associated with childlessness among partnered individuals. Some of these diseases are more likely to exert a direct biological effect on fertility or pregnancy complications (as in the case of diabetes and obesity), whereas others (such as some mental-health disorders) might impact family stability or delay the age at family formation. Additionally, in terms of risks for childlessness-associated early-life diseases, childless individuals are more similar to their siblings with one child than to those with higher parities. This suggests that the disease–childlessness associations identified from our main analyses might also inform the relationship between parental health status and low parity (that is, one child).

Our study has several strengths. First, for the majority of the study participants, we have almost complete coverage of health and reproductive information until the end of their reproductive period. Second, we considered more than 400 diseases across both men and women, allowing us to compare relative effects on childlessness both between diseases and between sexes. The use of nationwide data from two countries provided a large sample of 2.5 million and, most importantly, allowed us to assess how robust our findings were in different health-care systems and diagnostic practices. Finland has had a marked increase in childlessness since the 1970s of up to 22% for women and 32% for men[3,4], and our analyses reveal potential underlying mechanisms such as the wider detrimental effects of alcohol dependence in men and of endocrine–nutritional–metabolic disorders.

Third, by leveraging the information about family formation and singlehood, we could explore to what extent this information, which has been described as an important mediator of childlessness[50], mediated the association between disease and childlessness. Fourth, we used a matched-sibling design to limit confounding from a shared familial environment. In our study, individuals with siblings had only a slightly lower proportion of childlessness than the general population. Sibling designs have been extensively used in analyses of Nordic registers and have several advantages over population-based designs[51,52].

Our study also reveals the legacy of a troubled and unjust past of reproductive rights, often stemming from eugenic thinking and discrimination related to disability and gender[39,53,54]. We found that intellectual disability was the condition with the strongest association in men, but also high levels of childlessness were associated with cerebral palsy and behavioural and personality disorders. Until 1970 in Finland and 1976 in Sweden, individuals categorized as having severe mental disorders or physical disabilities—particularly women—were subjected to involuntary sterilization[39].

Our study also has several limitations. First, disease diagnoses were largely obtained from registers covering secondary and tertiary health care in hospitals that capture more severe disease cases (for example, mental health and behavioural conditions), and we thus lack information on the onset and occurrence of major diseases included in primary health-care data (for example, low-grade metabolic conditions). Moreover, disease diagnoses have changed over time; we have defined disease outcomes by harmonizing three versions of International Classification of Diseases (ICD) codes that might capture diseases with different accuracy. Our findings are based on individuals born between 1956 and 1973, and the results may thus not be entirely generalizable to more recent cohorts, because reproductive and partnering practices have changed and because better treatments might alleviate the effects of some diseases on childlessness.

Second, as with previous studies to date, we were unable to partition the impacts of diseases into voluntary and involuntary childlessness due to a lack of data on reproductive preferences and intentions and difficulties in differentiating the two[19,22,55]. Given that estimates of the voluntarily childfree group are around a quarter of the total population of people without children, we note that public health interventions should take this proportion into account and target those who are involuntarily childless due to certain diseases. Among factors that impact involuntary childlessness, long-term institutionalization and disease-related treatments are two aspects that future registry-based studies might be able to explore. For example, medications for epilepsy are known to be associated with pregnancy complications[56].

Third, assortative mating may lead to an overestimation of a disease's association with childlessness among partnered couples. Fourth, despite our attempt to use a study design that minimizes confounding and reverse causality, these biases might challenge the interpretation of the results. For example, although full siblings share roughly half of their genetic material and similar family-of-origin characteristics, and although only siblings with the closest birth order were considered, siblings can still experience different childhood conditions and even be influenced by each other's conditions[57]. In a sensitivity analysis, we adjusted for differences in siblings' highest education level and observed similar results compared to the main analyses; however, we note that adjusting for different socio-economic factors might affect the within-sibling estimates. Also, the use of only individuals with same-sex full siblings may introduce selection bias, although for most disease diagnoses, their effects obtained from sibling-based analyses were not significantly different from those obtained from population-based analyses.

Finally, there is the question of the generalizability of our results to other nations. The Nordic countries have been the forerunners of demographic change in the realm of partnerships and fertility[48,58], making these results relevant for other industrialized nations. For example, being married or having a registered partnership is not a precondition for having children in Nordic countries (for example, 14% of Finns with children never entered any marriages or registered partnerships), which is a pattern gradually becoming more prevalent in other nations. There are also important socio-economic fertility gradients in Finland and Sweden that mirror changes in many industrialized nations. In the Nordic countries, there has been a positive association between men's higher education and fertility[49,59]. For Nordic women, there was initially the reverse gradient (that is, higher levels of childlessness among those with higher education), but this has shifted over time, with women becoming increasingly similar to men[59]. This 'new Nordic' fertility regime links higher socio-economic status to higher fertility[60] but also to multi-partner fertility[61], a trend that has emerged in other nations as well.

In conclusion, we have comprehensively described the associations between different diseases, particularly those with onset prior to the peak reproductive age, and the chance of being childless over a lifetime. This evidence can be used as a basis for future studies focusing on prioritizing health interventions to counter involuntary childlessness.

## Methods

This study used nationwide registers from Finland and Sweden. The use of Finnish registry data is approved by the Digital and Population Data Service Agency (VRK/6551/2019-1 and VRK/6551/2019-2), Statistics Finland (TK-53-1813-19) and the Finnish Institute for Health and Welfare (THL/804/5.05.00/2019). The use of Swedish registry data is approved by Socialstyrelsen (27035/2018) and Statistics Sweden (247849). The Ethics Committee/IRB of Regional Ethical Review Board in Uppsala gave ethical approval for this work (2018/223).

### Study population

We considered all individuals born in Finland and Sweden between 1956 and 1968 (men) or between 1956 and 1973 (women) as index persons, for whom the vast majority completed their reproductive lifespan (45 years old for women[32,62,63] and 50 for men[32,63]) by 31 December 2018 (Fig. 1a). Individuals who emigrated during the study period were excluded to avoid incomplete follow-up for disease diagnoses and reproductive information. We further excluded individuals who died before the age of 16 to eliminate the effect of diseases on pre-reproductive survival. In total, we considered 1,425,640 index women (572,518 Finns and 853,122 Swedes) and 1,119,380 men (463,410 Finns and 655,970 Swedes). For these index individuals, we also obtained information for parents, spouses, siblings and children for a total of 9,305,692 individuals (3,640,464 Finns and 5,665,228 Swedes).

### Childlessness (main outcome)

The primary reproductive outcome of this study is childlessness, defined as having no live-born children by age 45 for women[32,62,63] and age 50 for men[32,63]. For every index person, we extracted demographic information on all biological children born before 31 December 2018 in Finland from the Population Information System and in Sweden from the Multi-Generation Register. Medical birth information including stillbirth and the use of assisted reproductive techniques was obtained from the Medical Birth Register, available since 1987 in Finland and 1973 in Sweden. We excluded children conceived by assisted reproductive techniques (0.3% in Finland and 0.8% in Sweden) to control for potential confounding from social inequalities in medical help-seeking for infertility, especially during the observational period.

### Marriage and partnership (secondary outcome)

For index individuals in Finland, we obtained their longitudinal marriage and partnership information between 11 April 1971 and 31 December 2018 from the Population Information System. In Sweden, we collected information on married couples and cohabiting unions with biological children between 1 January 1977 and 31 December 2017 from Statistics Sweden. We defined individuals as partnerless if they did not have any abovementioned marriage or partnership registered by age 45 (women) or 50 (men). We note that in Finland and Sweden, around 14% of individuals have children without being married or having a registered partnership (Supplementary Table 3) and that a registered partnership is rare among individuals with partners.

### Disease diagnosis (exposure)

A wide variety of disease endpoints were defined by clinical expert groups[64], through ICD codes of versions 8 (1969–1986), 9 (1987–1995 in Finland and 1987–1996 in Sweden) and 10 (1996–2018 in Finland and 1997–2018 in Sweden). We collected main diagnoses (ICD codes) and admission dates for secondary health-care inpatient hospital treatments (since 1969, but in Sweden, nationwide coverage began in 1973 for psychiatric diagnoses and 1987 for others[65]) and specialist outpatient visits in Finland from the Care Register for Health Care (since 1998) and in Sweden from the National Patient Register (since 2001). In Finland, we obtained cancer diagnoses (International Classification of Diseases for Oncology, 3rd edition) and corresponding dates of diagnoses from the Finnish Cancer Registry, available since 1953.

Additionally, for all individuals who died before the end of follow-up, we collected the date of death and basic, immediate and contributing causes of death from Statistics Finland and Statistics Sweden to capture additional disease diagnoses that were missing from hospital inpatient and outpatient registers. The age of onset was defined as the first record of a disease diagnosis.

In total, we considered 16 disease categories, with infectious–parasitic diseases and congenital anomalies defined only in Finland and all remaining categories defined in both countries. To remove highly correlated disease diagnoses, we estimated tetrachoric correlations between every two disease endpoints using individual-level data and only kept the one with the highest prevalence if a group of disease diagnoses had tetrachoric correlations higher than 0.7. After removing highly correlated disease diagnoses, we considered 414 diseases for which we had more than 30 affected individuals in the sibling-based analysis for each sex, in Finland or Sweden.

### Sibling-based analysis (main analysis)
There were 274,205 pairs of full sisters (118,978 in Finland and 155,227 in Sweden) and 212,849 pairs of full brothers (97,849 in Finland and 115,000 in Sweden) among index persons (Fig. 1a), which allowed us to control for potential confounding from genetic and environmental factors by performing a matched-pair case–control study design (Fig. 1b). First, we kept only families having at least two same-sex full siblings discordant for childlessness (at least one sibling with children and one childless). Then, within each family, we randomly selected one childless sibling as a case, and, as a control, the sibling with children that was the closest in birth order to the case. Childless siblings were of similar age to siblings with children (absolute mean age difference, 0.3 years (s.d. = 4.0) and 0.5 years (4.4) for full sisters and full brothers, respectively). Finally, within each sibling pair, for both the case and the control, we only considered disease diagnoses that had their onset at least one year before the age at first birth for the sibling with children, to eliminate the effect of diseases triggered by pregnancy. With the matched design, we excluded diseases diagnosed after the reproductive lifespan to avoid reverse causation. We used a conditional logistic regression model to investigate the association between a disease and childlessness by applying the clogit function implemented in the survival v.3.2-0 package[66]. Full siblings were likely to share environmental factors, especially during the pre-reproductive period. We therefore only adjusted for birth year effects. We used a similar model to calculate the association between diseases and singlehood. We applied Bonferroni correction ($P = 0.05/328 = 1.5 \times 10^{-4}$ for women and $P = 0.05/325 = 1.5 \times 10^{-4}$ for men, where 328 and 325 are the numbers of unique diseases considered in women and men, respectively) to control for the familywise error rate. The meta-analysis between estimates obtained from the two populations was conducted with the metagen function implemented in the meta v.5.1-0 package[67]. For all statistical analyses involved in this study, we used R v.4.1.2[68].

Additionally, for disease diagnoses that were significantly associated with childlessness, we assessed whether there were any age-of-onset-dependent effects by considering the age of onset (eight groups (≤15, 16–20, 21–25, 26–30, 31–35, 36–40, 41–45 or unaffected) for both sexes and an additional group (46–50) for men) as fixed effects. For individuals with children, we further assessed whether the effects of disease diagnoses were consistent across parities by comparing childless individuals to their siblings with one child, two children or more than two children.

### Mediation effect by singlehood
To assess the role that partnership formation plays in mediating the observed associations, we performed a causal mediation analysis[69] to partition the association between disease diagnosis and childlessness into indirect (mediated by singlehood) and direct components (Supplementary Information).

### Population-based analysis (secondary analysis)
To estimate the population-level effects of diseases on childlessness, we performed a nested incident-matched case–control design that matched each childless case to one control on the basis of sex, birth year, municipality of birth (545 from Finland and 1,091 from Sweden) and the highest parental education level (International Standard Classification of Education 1997) (Supplementary Information). Parental education level was used instead of the index person's education level to avoid confounding between an individual's educational attainment and medical conditions[33]. We used conditional logistic regression analysis with no additional covariates in the model. In total, we used 226,860 matched pairs of women (107,375 in Finland and 119,485 in Sweden) and 274,941 of men (127,689 in Finland and 147,252 in Sweden) for the population-based analysis.

### Sensitivity analysis
First, to eliminate the effect of diseases on reproductive-age mortality, we used the sibling design framework described above but restricted the analysis to 66,255 full-sister pairs (33,556 in Finland and 32,699 in Sweden) alive by age 45 and 66,009 full-brother pairs (32,996 in Finland and 33,013 in Sweden) alive by age 50. Second, we used the sibling design framework described above, but instead of conditional logistic regression, we performed a stratified Cox proportional hazards regression model considering chronological age as the time scale and disease status as a time (age) varying exposure (unaffected until disease onset and affected afterwards). Follow-up started at age 16, the estimated start of the reproductive lifespan, and was censored at death or one year before the age at first birth of the control, whichever occurred first. Third, to quantify to what extent the identified associations between diseases and childlessness were mediated by social characteristics such as educational attainment, we further adjusted for individuals' highest education level for both the sibling- and population-based analyses. All sensitivity analyses using conditional logistic regression were conducted with the clogit function implemented in the survival v.3.2-13 package, while the analyses using the Cox proportional hazards regression model were conducted with the coxph function implemented in the same package.

### Reporting summary
Further information on research design is available in the Nature Portfolio Reporting Summary linked to this article.

## Data availability
All results (aggregated data) can be explored on the interactive online dashboard available at https://dsgelrs.shinyapps.io/DiseaseSpecifi-cLRS/. Due to data protection regulations, we are not allowed to share individual-level data directly. However, all Finnish and Swedish register data used in this study can be applied from national data agencies including Statistics Finland (https://www.stat.fi/index_en.html), the Population Information System (https://dvv.fi/en/individuals), the Finnish Institute for Health and Welfare (https://thl.fi/en/web/thlfi-en/statistics-and-data/data-and-services/register-descriptions/care-register-for-health-care) and the Finnish Cancer Registry (https://cancerregistry.fi/) from Finland; and Statistics Sweden (https://www.scb.se/en/) and the National Board of Health and Welfare (Socialstyrelsen, https://www.socialstyrelsen.se/en/) from Sweden.

## Code availability
The code for conducting this study is available at https://github.com/dsgelab/Lifetime-Reproductive-Success.

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

## Acknowledgements

This work was supported by funding from the European Research Council (grant no. 945733) received by A.G., from the European Research Council (grant no. 835079), the Leverhulme Trust, Leverhulme Centre for Demographic Science (grant no. RC-2018-003) and Economic and Social Research Council/UKRI (grant no. ES/W002116/1) received by M.C.M., and from the Academy of Finland (grant no. 350399) received by S.L. The funders had no role in study design, data collection and analysis, decision to publish or preparation of the manuscript. The computations for the Swedish data were enabled by resources in project sens2019018 provided by the Swedish National Infrastructure for Computing at UPPMAX, partially funded by the Swedish Research Council through grant agreement no. 2018-05973. We thank T. Andersson, A. Sariaslan and G. Su for valuable advice on statistical modelling; G. Mignogna and D. Brazel for useful discussions during project planning; and V. Llorens for suggestions on disease coding and applying Cox proportional hazards regression.

## Author contributions

A.L., M.C.M. and A.G. designed the study and wrote the manuscript. A.L. conducted the statistical analysis and generated the figures and tables. E.T.A. and X.D. helped in performing the sensitivity analyses, assisted in interpreting the results from a sociodemographic standpoint, drafted Supplementary Fig. 1 and prepared Supplementary Table 1. S.J., P.V. and T.K. assisted in interpreting the results from a clinical standpoint. T.K. and A.S.H. defined the disease endpoints. S.K., A.G. and A.L. developed the R shiny application. T.F. and M.G. coordinated the data application and extraction for Swedish and Finnish nationwide registers. S.L. contributed to the editing of socio-economic data and statistical modelling. All authors discussed the results, revised the manuscript and had final responsibility for the decision to submit for publication.

## Competing interests

The authors declare no competing interests.

## Additional information

**Correspondence and requests for materials** should be addressed to Aoxing Liu, Melinda C. Mills or Andrea Ganna.

# Reporting Summary

## Statistics

For all statistical analyses, confirm that the following items are present in the figure legend, table legend, main text, or Methods section.

| n/a | Confirmed | |
|---|---|---|
| ☐ | ☒ | The exact sample size (*n*) for each experimental group/condition, given as a discrete number and unit of measurement |
| ☐ | ☒ | A statement on whether measurements were taken from distinct samples or whether the same sample was measured repeatedly |
| ☐ | ☒ | The statistical test(s) used AND whether they are one- or two-sided *Only common tests should be described solely by name; describe more complex techniques in the Methods section.* |
| ☐ | ☒ | A description of all covariates tested |
| ☐ | ☒ | A description of any assumptions or corrections, such as tests of normality and adjustment for multiple comparisons |
| ☐ | ☒ | A full description of the statistical parameters including central tendency (e.g. means) or other basic estimates (e.g. regression coefficient) AND variation (e.g. standard deviation) or associated estimates of uncertainty (e.g. confidence intervals) |
| ☐ | ☒ | For null hypothesis testing, the test statistic (e.g. *F*, *t*, *r*) with confidence intervals, effect sizes, degrees of freedom and *P* value noted *Give P values as exact values whenever suitable.* |
| ☒ | ☐ | For Bayesian analysis, information on the choice of priors and Markov chain Monte Carlo settings |
| ☒ | ☐ | For hierarchical and complex designs, identification of the appropriate level for tests and full reporting of outcomes |
| ☒ | ☐ | Estimates of effect sizes (e.g. Cohen's *d*, Pearson's *r*), indicating how they were calculated |

*Our web collection on statistics for biologists contains articles on many of the points above.*

## Software and code

Policy information about availability of computer code

| Data collection | No code was used to collect data in the study. |
|---|---|
| Data analysis | For the sibling-based analysis and population-bases analysis, we conducted a conditional logistic regression using the clogit function of the survival_3.2-13 package. For the sensitivity analysis using a time-varying cox PH model, We used the coxph function of the survival_3.2-13 package. When meta-analyzing estimates from two populations, we used the metagen function of the meta_5.1-0 package. For the mediation analysis, we used the CDMA and the VAR.D developed by [PMID: 32608110]. For all above statistical analyses, we used R version 4.1.2. |

For manuscripts utilizing custom algorithms or software that are central to the research but not yet described in published literature, software must be made available to editors and reviewers. We strongly encourage code deposition in a community repository (e.g. GitHub). See the Nature Portfolio guidelines for submitting code & software for further information.

## Data

Policy information about <u>availability of data</u>

All manuscripts must include a <u>data availability statement</u>. This statement should provide the following information, where applicable:

- Accession codes, unique identifiers, or web links for publicly available datasets
- A description of any restrictions on data availability
- For clinical datasets or third party data, please ensure that the statement adheres to our <u>policy</u>

Data availability statement:
Due to data protection regulations, we are not allowed to share individual-level data directly. However, all Finnish and Swedish register data used in this study can be applied from national data agencies. All Finnish and Swedish register data used in this study can be applied from national data agencies including Statistics Finland (https://www.stat.fi/index_en.html), Population Information System (DVV, https://dvv.fi/en/individuals), Finnish Institute for Health and Welfare (THL, https://thl.fi/en/web/thlfi-en/statistics-and-data/data-and-services/register-descriptions/care-register-for-health-care) and Finnish Cancer Registry (https://cancerregistry.fi/) from Finland, and Statistics Sweden (https://www.scb.se/en/) and National Board of Health and Welfare (Socialstyrelsen, https://www.socialstyrelsen.se/en/) from Sweden.
All results (aggregated data) can be explored on the interactive online dashboard available at https://dsgelrs.shinyapps.io/DiseaseSpecificLRS/.

## Human research participants

Policy information about <u>studies involving human research participants and Sex and Gender in Research.</u>

| Reporting on sex and gender | We collected the information of sex for each participant from the nationwide population system. The analysis has been done separately for two sexes. Given that the previous studies mainly focus on women and lack of evidence for men, we further investigated the sex-difference for each identified association. |
|---|---|
| Population characteristics | We examined all individuals born in Finland (n=1,035,928) and Sweden (n=1,509,092) between 1956 and 1968 (men) or 1956 and 1973 (women) and followed them up until the end of 2018, when most have completed their reproductive lifespan. Socio-demographic, health, and reproductive information was obtained from nationwide registers. For the entire study population, we have information from nationwide registers covering 414 disease diagnoses across 16 main categories. |
| Recruitment | We considered all individuals born in Finland (n=1,035,928) and Sweden (n=1,509,092) between 1956 and 1968 (men) or 1956 and 1973 (women) and followed them up until the end of 2018. Therefore, no selection has been done in terms of participant recruitment. |
| Ethics oversight | Ethics committee/IRB of Regional Ethical Review Board in Uppsala gave ethical approval for this work (2018/223). The use of Swedish registry data for this study is approved by socialstyrelsen (permit number: 27035/2018) and Statistics Sweden (permit number: 247849). The use of Finnish registry data is approved by Digital and population data service agency (permit numbers: VRK/6551/2019-1 and VRK/6551/2019-2), Statistics Finland (permit number: TK-53-1813-19), and Finnish Institute for Health and Welfare (permit number: THL/804/5.05.00/2019). |

Note that full information on the approval of the study protocol must also be provided in the manuscript.

# Field-specific reporting

Please select the one below that is the best fit for your research. If you are not sure, read the appropriate sections before making your selection.

☒ Life sciences        ☐ Behavioural & social sciences        ☐ Ecological, evolutionary & environmental sciences

For a reference copy of the document with all sections, see nature.com/documents/nr-reporting-summary-flat.pdf

# Life sciences study design

All studies must disclose on these points even when the disclosure is negative.

| Sample size | We defined our main outcome, lifetime childlessness, as individuals that have had no live-born children by the end of their reproductive lifespan (age 45 for women; 50 for men). To have virtually complete coverage of health and reproductive information until the end of reproductive period, we examined all individuals born 1956-1968 (men) and 1956-1973 (women) in Finland (n=1,035,928) and Sweden (n=1,509,092) to completion of their reproductive lifespan in 2018 (age 45 for women and 50 for men). For these index individuals, we also obtained information for parents, spouses, siblings, and children for a total of 9,305,692 individuals (3,640,464 Finns and 5,665,228 Swedes). With the unique datasets, we were able to analyze 414 diseases for which we had more than 30 affected individuals in the sibling-based analysis for each sex, in Finland or Sweden. |
|---|---|
| Data exclusions | Individuals who emigrated during the study period were excluded to avoid incomplete follow-up for disease diagnoses and reproductive information. We further excluded individuals who died before the age of 16 to eliminate the impact of diseases on pre-reproductive survival. |

We excluded children conceived by assisted reproductive techniques (0.3% Finland, 0.8% Sweden) to control for potential confounding from social inequalities in medical help-seeking for infertility, especially during the observational period.

| | |
|---|---|
| Replication | The use of nationwide data from two countries provided a large sample of 2.5 million and allowed us to assess how robust our findings were to different healthcare systems and diagnostic practices. The identified associations with diseases such as type 1 diabetes and several major mental health disorders were consistent with the previous studies [PMID: 17563340; 23147713]. |
| Randomization | No randomization was performed because we considered everyone from Finland and Sweden born in certain years. |
| Blinding | Blinding was not relevant to the study because this was a population-based study, with all analyzed data obtained from nationwide registers. |

# Reporting for specific materials, systems and methods

We require information from authors about some types of materials, experimental systems and methods used in many studies. Here, indicate whether each material, system or method listed is relevant to your study. If you are not sure if a list item applies to your research, read the appropriate section before selecting a response.

## Materials & experimental systems

| n/a | Involved in the study |
|---|---|
| ☒ | ☐ Antibodies |
| ☒ | ☐ Eukaryotic cell lines |
| ☒ | ☐ Palaeontology and archaeology |
| ☒ | ☐ Animals and other organisms |
| ☒ | ☐ Clinical data |
| ☒ | ☐ Dual use research of concern |

## Methods

| n/a | Involved in the study |
|---|---|
| ☒ | ☐ ChIP-seq |
| ☒ | ☐ Flow cytometry |
| ☒ | ☐ MRI-based neuroimaging |

