## [Peer Review File · Nature Human Behaviour]

Peer Review Information

Journal: Nature Human Behaviour

Manuscript Title: Evidence from Finland and Sweden on the relationship between early-life diseases and lifetime childlessness in men and women

Corresponding author name(s): Aoxing Liu, Melinda C Mills, and Andrea Ganna

Reviewer Comments & Decisions:

Decision Letter, initial version:

9th October 2022

Dear Dr Ganna,

Thank you once again for your manuscript, entitled "The relationship of major diseases with childlessness", and for your patience during the peer review process.

Your Article has now been evaluated by 3 referees. You will see from their comments copied below that, although they find your work of considerable potential interest, they have raised quite substantial concerns. In light of these comments, we cannot accept the manuscript for publication, but would be interested in considering a revised version if you are willing and able to fully address reviewer and editorial concerns.

We hope you will find the referees' comments useful as you decide how to proceed. If you wish to submit a substantially revised manuscript, please bear in mind that we will be reluctant to approach the referees again in the absence of major revisions. We are committed to providing a fair and constructive peer-review process. Do not hesitate to contact us if there are specific requests from the reviewers that you believe are technically impossible or unlikely to yield a meaningful outcome.

To guide the scope of the revisions, the editors discuss the referee reports in detail within the team, including with the chief editor, with a view to (1) identifying key priorities that should be addressed in revision and (2) overruling referee requests that are deemed beyond the scope of the current study. We hope that you will find the prioritised set of referee points to be useful when revising your study. Please do not hesitate to get in touch if you would like to discuss these issues further.

In your revision, we ask that you include more granular analyses by the number of children (Reviewer #1) and analyses for indirect effects with confounders as suggested by Reviewer #2. Please be specific in your abstract/title regarding the context of the study (two Nordic countries) and please discuss generalizability of your results. We also ask that you remove suggestions that these results can be used for targeted health interventions for the results that are not causal, and also please remove all language that paints childlessness as a problem that needs to be addressed.

If you wish to submit a suitably revised manuscript we would hope to receive it within 4 months. I would be grateful if you could contact us as soon as possible if you foresee difficulties with meeting this target resubmission date.

- Include a "Response to the editors and reviewers" document detailing, point-by-point, how you addressed each editor and referee comment. If no action was taken to address a point, you must provide a compelling argument. When formatting this document, please respond to each reviewer comment individually, including the full text of the reviewer comment verbatim followed by your response to the individual point. This response will be used by the editors to evaluate your revision and sent back to the reviewers along with the revised manuscript.
- Highlight all changes made to your manuscript or provide us with a version that tracks changes.

[REDACTED]

Thank you for the opportunity to review your work. Please do not hesitate to contact me if you have any questions or would like to discuss the required revisions further.

Sincerely,

Arunas Radzvilavicius, PhD
Editor, Nature Human Behaviour
Nature Research

Reviewer expertise:

Reviewer #1: public health, epidemiology, demography

Reviewer #2: psychiatric epidemiology

Reviewer #3: public health, epidemiology, demography

REVIEWER COMMENTS:

Reviewer #1:

Remarks to the Author:

Thank you for giving me the opportunity to review this high quality paper, which explores the association between having no live births and diseases prior to reproduction, recorded in hospitalisation and specialist disease registers in Finland and Sweden.

The topic of fertility and health is an important one, which has been under investigation for some time - mainly among the demography/ social epidemiology fields. We already know that people with no biological children have numerous health disadvantages. There are many important citations missing, which have used cause of death data, surveys etc. However, this study is unique in its access to exceptionally rich medical data at a population level. The authors have conducted a number of advanced analyses, taking full advantage of the large sample size, that aim to reduce confounding effects and investigate mediation processes. Because of this exceptionally rich disease data does represent a step change in our understanding of the relationship between nulliparity and health. It is also very well written.

That said, I have several concerns with the paper. The first is that the paper might produce a negative impact, if it were published in its present form. The focus on childless vs parous (without any attention to the nuances of number of children, e.g. differences between 1, 2, 3, etc) might promote social stigmatisation of childless individuals. The authors easily have the sample size to be able to explore how childless individuals compare to those with high parity, for example.

The authors also imply that the relationship between childlessness and major diseases will be consistent in other contexts, but the Nordic countries they use are really quite unusual. First, they have relatively high fertility in the European context, and a very strong 2-child norm, and those that have fewer children than 2 children are socially selected in various ways. In settings where childlessness is more common, and as childlessness becomes more normative, these associations with disease are likely to reduce. The relationship between fertility and various other health outcomes has been shown to be moderated by social context/ social norms (underlined by the mediation by partnership status) , which the authors don't acknowledge. Rather, the article is written as though this set of associations reflects a universal (biological) pattern, rather than being partly the product of a specific set of social and cultural conditions. I am very surprised to not see a richer discussion of childlessness and health, within broader cohort patterns, inter generational processes of disadvantage, and social selection.

The last concern is that the analysis is not accurately represented in the title, abstract and interpretation. The main analysis - the sibling design used for the main analysis is very robust, in that it cuts the observation time at one year prior to first childbirth in the parous sibling. But what this means is that the study focuses on diseases occurring very early in the life course and are severe enough to require hospitalisation. This gives a very different perspective to considering all major diseases.

Therefore, although from a scientific point of view, the analysis is impressive, I think the framing/interpretation should be further nuanced, to avoid contributing to social stigma experienced by childless individuals, and to represent the evidence the study provides more accurately.

I also have the following detailed comments/ suggestions:

Abstract

The first sentence of the abstract is extremely vague. What countries in the world are these percentages based on? Globally? High income countries? Is childlessness increasing everywhere?

I think the abstract misses an explanation of what we already know, which is considerable. There have been many studies based a) cause of death data b) prescription data / disease registers c) survey data already. We already know that nulliparous individuals are more likely to have certain types of health behaviours and health conditions. What is missing - and this study provides- is a more comprehensive picture going into detail of these diseases prior to childbearing.

Introduction

I find the introduction presents the issues a bit simplistically and does little to place childlessness in social and cultural contexts.

Again, the first sentence needs some context- I see you reference Sobotka, but which countries exactly, and what time period- needs to be more specific. The second sentence uses data on nulliparity at age 40- some of this is related to postponement.

I really object to the painting of childlessness as a problem that needs to be addressed- why does it present a challenge for societal sustainability? Some people would really argue the opposite (not that I support such polarised perspectives, either). I think this is done to develop the motivation for the paper but this could be done in a more nuanced way.

I am very happy to see a conceptual framework which describes the possible mediators between disease and childless status, but again its a bit simplistic. What is the role of intergenerational health/fertility preferences, for example? Could it be the case that challenges conceiving could affect health behaviours and then disease status, in turn? Then, for the comparison group, who have children, depending on the number of timing, this affects their health behaviours and disease profiles, so it makes the childless individuals seems more/less likely to suffer from some conditions.

The third paragraph- if its intended to be a literature review of what is already know, ignores existing (more demographically) oriented studies which use cause of death data, survey data etc, which already point to the relationship between parity and various diseases (CVD, cancer, alcohol use etc).

Methods

The Figure is a very neat way of describing the analysis sample. In the text, it is not clear to me why

you include individuals across three generations. I think you could include mention of how you select siblings within same-sex sibling groups of more than 2. I also think the fact that you drop 80% of the sample should be reflected on- how are these individuals different?

One aspect which I think deserves better representation in the paper. So, if in the sibling analysis, you cut the observation time one year before the parous sibling has children, then what is the mean age of cutting the time? This basically means that you are observing diseases that occur only in really young people. So, the statements later on about observing childless individuals over their reproductive lifespan is only true for the robustness checks. The main analysis - which most of the paper is based on - is on diseases that occur mainly before aged 30? If so, I think the title, abstract and interpretation of data need to adjusted to make this clearer.

One factor which is not mentioned. You combine the data for these two countries: whats the justification for using both? why bother, especially when the disease data are sometimes a little different? Linked to the above, is this a sample size issue? Given that these are quite young people, are some of these diseases quite rare?

Marriage and partnership

How does the data account for cohabitating unions? I am not sure but I think for at least some of that period in Sweden data are available only for marriages.

Disease data

If the cancer registry is only available for Finland, how do they Swedish and Finnish cancer rates compare? Even if disease prevalence for cancer is similar, I would assume that Finns are 'diagnosed' at an early stage, because Swedes get their cancer data from hospitalisation ? Or does it not vary? If it does, I would be tempted to keep the disease data identification identical, or to justify better why its ok to keep it different.

Design

I think you overstate the ability of sibling FE to control for confounding. Siblings are not that similar. Siblings share 50% genetic material, similar family of origin characteristics, but can experience quite different childhood conditions. For example, birth order and family size for the majority of their time can result in quite different health profiles. As you adjust for year of birth, this takes account of who is older and by how much, but that's it, really. What about the other siblings in the family? Parental status - marriage- divorced and the age at which this occurred?

Results

Given the conceptual framework you include, I think it would make sense to have a more detailed set of descriptives which describe the social characteristics of the childless/controls in the two countries. I don't see anything which describes the education levels of the parents, health patterns during childhood, partnership patterns, parity, social class, education, of the cases and controls.

Fig 2: You need to include CIs on the main left hand side graphs. Some of the ORs for individual conditions are enormous but I'm assuming they also have very large CIs, as they are quite rare. It's also not very clear what the labels on the charts (left hand side) refer to to, I think you need some lines to link the labels to the correct marker.

I find that the analysis of # children unborn etc, although allows some interesting statements to be made, is not very useful, because it assumes that the disease led to the childlessness, rather than be the other way around or be mutually reinforcing.

Discussion

I would appreciate in the discussion, some more nuanced reflection on unique context (as discussed above), beyond the generic statement that it should be replicated in other populations. Further, as voluntary childlessness becomes a more important component in future, these results might lose their relevance. The limitations could be more detailed. The study presents this data, which does not include primary care data, to be representative of the onset and occurrence of major diseases. Some diseases are very unlikely to be well covered and the bias will be for capturing more severe aspects of mental health and behavioural conditions. Others such as low grade metabolic conditions may never be captured using these data sources.

The final statement- about the public health impact- could be further developed, based on the

acknowledgement that the study focuses on early life diseases.

Reviewer #2:

Remarks to the Author:

This manuscript sought to assess the relationship of disease to childlessness, across two national (Scandinavian) cohorts. Strengths of the study include the population-level data, the breadth of diseases considered (and the attendant methods used to reign in their numbers for informative results), and the care taken to assess associations by gender.

My primary concerns with the manuscript are as follows:

1. I wonder about the choice of the cohort (birth age 1956-1973); it seems ideal to include the widest cohort possible, particularly given changes in diagnostic processes over time (e.g., might diagnoses be coming earlier, generally, for instance?). Individuals born into the 80s are at or near 40 at present, and would offer an expanded group of individuals with more current medical histories who has passed through a substantial portion of child-bearing/rearing years. This would seem particularly import for the mental/behavioral disorders found to be central drivers in the analyses? Related: are there cohort effects that could be examined here (beyond sensitivity analyses)controlling for age at diagnosis; as our understanding of the burden of disease has advanced, it stands to reason changes have occurred in the relationship of disease to childlessness?

2. The manuscripts makes clear its intention to provide evidence which 'can be used for targeted health interventions to counter decreasing fertility, reproductive health, involuntary childlessness, and shrinking populations.' However, the methods do not offer any means of assessing whether - and to what degree - the childlessness of its disease-affected populations are attributable to meaningfully targetable reasons (indeed, characterizing even 'voluntary' reasons as changeable dismisses the economic and personal realities that shape the 'choice'). This is a non-trivial issue, as there are many diseases for which pregnancy is counter-indicated (whether due to dangers posed to the mother/child-bearer, or risk for offspring health), which pose direct, biological challenges to reproduction, or whose costs limit the feasibility of child-rearing.

The authors acknowledge this, to an extent, in stating that 'there are multiple non-disease related reasons for individuals to postpone parenthood or not have children ... individual choice, work-life reconciliation, housing and economic uncertainty, or inability find a partner.' However, no attempt seems to be made to integrate the majority of these features into the analysis or, at any length/depth, in their discussion. Certainly, data relevant to these features are available in the Scandinavian registers (e.g., education, unemployment histories?) - are the authors able to integrate? Given the long (and, indeed, lengthening) timespan constituting the 'child bearing/rearing' years, use of same-sex siblings (who will only share early-life exposures) does not address the likely differential environments characterizing the affected vs. unaffected siblings through their reproductive adulthoods. Likewise, such information would be a valuable covariate in the sensitivity analyses abandoning the sibling structure.

3. The results indicated that mental-behavior disorders and congenital anomalies had the strongest relationship with childlessness. However, congenital anomalies - per the supplement - could only be assessed in the Finnish population. Could the authors shed light on the reasons for this? How should we interpret the result, given the differences in rates of childlessness underlying the two national populations?

4. I am left to wonder about the impact of having an ill sibling. There is substantial literature highlighting the impact one ill child can pose on the family unit as a whole (e.g., in terms of caretaking responsibilities, availability of parental support, familial economic stability, likelihood of divorce, etc. - all features which may factor into siblings' own views on childlessness). There is, further, evidence to suggest that female children are more likely to be given caretaking and other responsibilities in these scenarios, regardless of whether the affected sibling is male or female. It would be interesting to see a familial analysis which included all siblings and/or permitted dissection of associations across gender (e.g., to see whether having an affected male sibling, for instance, had an association with childlessness among unaffected female siblings).

5. The discussion is severely lacking in references to support key statements, and even full paragraphs? Further, there are interpretive statements in the results that should be necessitate inclusion (and expansion) in the discussion (e.g., the role of sterilization, the reality of reproductive danger in certain disorders).

Smaller Items:

Line 135 missing 'and'

-

Fig 3C. The legend obscures the CIs/elder data points for Obesity

Also in 3C: I wonder about the selection of Anorexia here, given there are insufficient figures to conduct the male estimates?

-

Line 179: I question the assertion that malformations of the digestive and musculoskeletal system can be said to be meaningfully 'more strongly associated with childlessness in women than men,' given the overlapping CIs for the former (and the near overlap on the latter).

-

Line 188: Again, the CIs overlap for diabetes; and while the interpretation is important, perhaps this belongs in the discussion alongside a wider discussion of the differential causes underlying the signaled associations a various diseases with childlessness.

-

Line 237: what does 'TFR' mean? Is it defined in text?

Reviewer #3:

Remarks to the Author:

Key results: Please summarise what you consider to be the outstanding features of the work.

In this study, the authors use comprehensive Finnish and Swedish registry data to examine the relationship between major diseases and childlessness. Using a twin design, the authors show that various major diseases are indeed related to childlessness. For some diseases, their findings echo previous studies, but for other major diseases, this paper is the first to show a relationship with childlessness. The authors have included several valuable additional analyses, e.g. on the age of diagnosis. All in all, I consider this paper a key contribution to the literature on the relationship between health and childlessness.

Validity: Does the manuscript have flaws which should prohibit its publication? If so, please provide details.

No, this is a meticulously planned, executed and written up study, and I could not identify any flaws that would limit the paper's contribution to the literature or invalidate its key conclusions.

Originality and significance: What are the major claims of the paper? Do you think that they represent a significant advance in the field? If the conclusions are not original, please provide relevant references. On a more subjective note, do you feel that the results presented are of immediate interest to many people in your own discipline, and/or to people from several disciplines?

The main claims of the paper are around the association between various major diseases and childlessness. I see the contributions of this paper mostly in terms of evidence-based advance; conceptually, the link between major diseases (and health more generally) and childlessness is not new, and the twin design that has been used here has also already been applied in earlier work. Nonetheless, I would say that the evidence-based advance is significant, especially because the authors have not just confirmed associations between diseases and childlessness that were already established in earlier work, but have also established relationships between diseases and childlessness that were not identified yet in previous research. Moreover, given the combination of comprehensive registry data and the twin design, the relationships as established here are more robust than most earlier work.

Data & methodology: Please comment on the validity of the approach, quality of the data and quality

of presentation. Please note that we expect our reviewers to review all data, including any extended data and supplementary information. Is the reporting of data and methodology sufficiently detailed and transparent to enable reproducing the results?

The comprehensive registry data are of excellent quality, the data are presented clearly, and the reporting of the data and methodology is sufficiently detailed and transparent to allow reproducing the findings.

Preregistration: If any part of the work reported in the manuscript was pre-registered, did the authors follow their preregistration plan? Did they report any deviations from their preregistration? Note that we ask authors to provide a link to the pre-registration in the Methods section and state the date of pre-registration. We also ask that authors disclose all deviations from the pre-registered protocol and explain the rationale for deviation (e.g., flaw, suboptimality, or reviewer/editorial request). In cases of deviation from the analysis plan, the originally planned analyses need to be reported in Supplementary Information.

Not applicable

Appropriate use of statistics and treatment of uncertainties: Please include in your report a specific comment on the appropriateness of any statistical tests, and the accuracy of the description of any error bars and probability values.

The statistical tests used are all appropriate given the data and research questions, and issues around error and statistical significance are dealt with sufficiently.

Custom code: If the work includes custom code, does the code run as intended? If you are unable to access the code, please contact us.

Not applicable.

Conclusions: Do you find that the conclusions and data interpretation are robust, valid and reliable? Yes, the conclusions are supported by the results, and the presentation and interpretation of the data and the findings support the robustness, validity and reliability of the data.

Suggested improvements: Please list additional analyses, experiments or data that could help strengthening the work in a revision.

I have only one suggested improvement of the data. Given the focus on people with siblings, I wondered to what extent the conclusions can be generalized to the population level (after all, not having siblings may also indicate e.g. hereditary issues with fertility). The addition of the population-based analyses in the paper have significantly reassured me that this issue does not affect the main conclusions. However, I would recommend adding a few lines to the results and/or discussion section that reflect on to what extent some specific results (e.g. the estimates of the numbers of people who remained childless because of major diseases) might be slightly biased due to the focus on people with siblings in the main analyses.

References: Does this manuscript reference previous literature appropriately? If not, what references should be included or excluded?

Yes, to me the manuscript does justice to the existing literature on this topic.

Clarity and context: Is the abstract clear, accessible? Are abstract, introduction and conclusions appropriate?

Yes, the abstract is clear and easy to follow, and the abstract, introduction and conclusions are appropriate in light of the approach and findings of the study.

Please indicate any particular part of the manuscript, data, or analyses that you feel is outside the scope of your expertise, or that you were unable to assess fully.

Not applicable.

REVIEWER COMMENTS:

Reviewer #1:

1. Thank you for giving me the opportunity to review this high quality paper, which explores the association between having no live births and diseases prior to reproduction, recorded in hospitalisation and specialist disease registers in Finland and Sweden. The topic of fertility and health is an important one, which has been under investigation for some time - mainly among the demography/social epidemiology fields. We already know that people with no biological children have numerous health disadvantages. There are many important citations missing, which have used cause of death data, surveys etc. However, this study is unique in its access to exceptionally rich medical data at a population level. The authors have conducted a number of advanced analyses, taking full advantage of the large sample size, that aim to reduce confounding effects and investigate mediation processes. Because of this exceptionally rich disease data does represent a step change in our understanding of the relationship between nulliparity and health. It is also very well written.

That said, I have several concerns with the paper. The first is that the paper might produce a negative impact, if it were published in its present form. The focus on childless vs parous (without any attention to the nuances of number of children, e.g., differences between 1, 2, 3, etc) might promote social stigmatisation of childless individuals. The authors easily have the sample size to be able to explore how childless individuals compare to those with high parity, for example.

Answer: We thank the reviewer for these insightful comments.

First, we have entirely rewritten the introductory sections and added new material in the Supplementary Material section to provide a:

- (a) much more extensive review of the demographic literature on the reasons and consequences of childlessness; and,
- (b) added a new section on previous literature that have examined the relationships of childlessness and diseases, which is accompanied by
- (c) a new Table and section in the Supplementary Material providing an overview of additional citations related to cause of death, epidemiological, and survey data findings.

Second, in addition to the more detailed, extensively cited literature noted above, we have substantially revised the language throughout the paper to avoid any stigmatisation of childless individuals. Given that this is an essential element to understand the results and to clarify our position, we have now added a Text Box that defines childlessness and clarifies the literature, public discussion and potential for stigmatisation.

On Manuscript p 3.

“Text box: Defining Childlessness

We use the term childlessness to describe individuals that have had no live-born children by the end of their reproductive lifespan (age 45 for women; 50 for men). Childlessness is defined in the literature as being both involuntary, related to biology and fecundity (e.g., infertility, inability to find a partner) and voluntary or ‘childfree’¹ (e.g., active choice, preference²). It has been estimated that 4-5% of the current 15-20% women who are childless in Europe are voluntary childless³. Childless individuals are subjected to discrimination and marginalization in many societies⁴, with infertile women globally experiencing multiple types of violence and coercion⁵. A parallel line of work, which is not the position of this paper or authors, is to problematize and stigmatize childless individuals as egoistic and place blame on this group for producing a so-called ‘demographic disaster’ of shrinking and ageing

populations and collapse of social security systems⁶. The approach of this paper is to provide a neutral, data-driven, and factual examination of early-life diseases related to childlessness, with the aim to design a better understanding of health to prevent childlessness among those who want to have children.”

As noted above, we have now also completely rewritten the **Introduction** and **Abstract** with a more extensive and balanced literature review and removed any text that might inadvertently stigmatise childless individuals.

The updated Abstract (**Manuscript p 2**):

“The percentage of women born 1965-1975 remaining childless is ~20% in many Western European and ~30% in some East Asian countries. Around a quarter of childless women do that voluntarily, suggesting a remaining role for disease. Single diseases have been linked to childlessness, mostly in women, yet we lack a comprehensive picture of the effect of early-life diseases on lifetime childlessness. We examined all individuals born 1956-1968 (men) and 1956-1973 (women) in Finland (n=1,035,928) and Sweden (n=1,509,092) to completion of reproduction in 2018 (age 45 women; 50 men). Leveraging nationwide registers, we associated sociodemographic and reproductive information with 414 diseases across 16 categories, using a population and matched pair case-control design of siblings discordant for childlessness (71,524 full-sisters, 77,622 full-brothers). The strongest associations were mental-behavioural, particularly amongst men (schizophrenia, acute alcohol intoxication), congenital anomalies and endocrine-nutritional-metabolic disorders (diabetes), strongest amongst women. We identified novel associations for inflammatory (e.g., myocarditis) and autoimmune diseases (e.g., juvenile idiopathic arthritis). Associations were dependent on age at onset, earlier in women (21-25 years) than men (26-30 years). Disease association was mediated by singlehood, especially in men and by educational level. Evidence can be used to understand how disease contributes to involuntary childlessness.”

The newly added introduction paragraphs (**Manuscript p 3-5**):

“In Western European countries, around 20% of women born around 1965 remained childless, with the highest levels of permanent childlessness now in East Asia, ranging from 28% (Japan) to 35% (Hong Kong) for women born in 1975^{7,8}. There has also been a sharp increase in childlessness over the last decades in countries such as in Finland, increasing since the 1970s from 14% to 22% (women aged 40 years) and 22% to 32% (men)^{9,10}. Demographic research has isolated core factors linked to the rise in contemporary levels of childlessness.

*Access to effective contraception at the end of the 1960s in many countries and more recently, emergency post-coital contraception in the late 1990s, provided couples with more abilities to control their reproduction⁷. Women’s opportunities also vastly changed in many industrialised nations since the 1980s, with gains in higher levels of education and entry into the labour market accompanied with shifts in gender norms and equity^{11,12}. At the same time, couples faced work-life reconciliation constraints when planning to have children including lack of access to childcare, challenging housing conditions, economic uncertainty, inability to find a partner, and the general absence of supportive family policies¹². Given that women entered the labour market while still taking on the bulk of household labour, this ‘incomplete gender revolution’ resulted in women often being forced to decide between a career and parenthood^{13,14}. Higher levels of job strain and lack of work-life reconciliation has also been found to reduce fertility intentions¹⁵. Another shift has been the growth of precarious, temporary, and uncertain employment, leading many to postpone or forgo entering partnerships and parenthood^{16,17}. Many of these changes resulted in a general postponement of children to later ages¹². This in turn meant that individuals were having children at ages when they had lower fecundity (reproductive capacity), leading to infertility-related issues and having fewer children¹⁸. This also resulted in an increase in involuntary childlessness (see **Text Box**).*

Although economic, societal, and cultural freedom has been linked to individuals who remained ‘childfree’ or voluntarily childless¹⁹, the individuals who report that they never intended to have

children, however, has remained low and is estimated to be around 5% in Europe³. Rather than being planned at an early age, becoming voluntarily childfree appears to be a mix of unforeseen circumstances, often related to postponement of childbearing and adaptation to a childfree life^{20,21}. This makes the decomposition of measuring voluntary versus involuntary childlessness challenging. A longitudinal study measuring fertility intentions of childless women (questioned from age 14 to their late 40s) found that they engaged in repeated postponement of childbearing and the subsequent adoption of a childless expectation at older ages or had indecision about parenthood evidenced by changing childless expectations across various ages.

The majority of childless individuals are thus involuntarily childless and wanted to have children. From the standpoint of public health, involuntary childlessness may impact other health domains, with childless women more likely to suffer from relationship dissolution, lower levels of self-esteem and isolation, and higher risks of clinical depression¹². A recent systematic review found that infertile women globally are also more likely to experience psychological, physical, and sexual violence as well as economic coercion⁵.

While the aforementioned demographic and socioeconomic literature has shown social, economic, and structural factors as having a major effect on childlessness, the fact that a large proportion of individuals are still involuntarily childless suggests that biology and diseases likely have a considerable influence on individuals' chances of being childless or having a particular number of children over their lifetime (**Figure S1**). First, diseases may directly influence childlessness through medical conditions that affect the fecundity, co-morbidities and mortality of the individual, as well as the risk of stillbirth²². Second, diseases can impact selection into a partnership, which in turn lowers the chance of being partnered and thus having children²³. Also, some may choose to be childfree due to knowledge or concerns of intergenerational transmission of certain known genetic conditions²⁴. Finally, many diseases are associated with lower socioeconomic status, unemployment, and economic uncertainty, which may amplify the psychosocial impact of diseases on childlessness²⁵.

A review of the literature linking disease to childlessness can be found in **Table S1**. Previous medical studies largely only examined a limited selection of diseases directly associated with reproductive biology such as recurrent miscarriage²⁶, polycystic ovary syndrome²⁷, and endometriosis²⁸. Another large area of research has been the study of infertility and inflammatory bowel disease (IBD). A systematic review found that up to one-third of IBD patients opted to be voluntarily childless, often related to lack of knowledge related to pregnancy-related IBD issues²⁹, with clinical studies finding that women with moderate to severe IBD had higher incidences of adverse pregnancy outcomes³⁰. Multiple sclerosis has also been associated with higher levels of childlessness, linked to fewer being in a stable relationship, fear of genetically transmitting multiple sclerosis, or discontinuation of treatments³¹. Another line of study has examined fecundity in patients with major psychiatric disorders³², and those with a genetic liability for schizophrenia, with the later finding no clear association with childlessness³³.

Third, we have conducted parity-stratified analyses suggested by the reviewer to study the effect of different diseases prior to reproduction not only on having 0 vs 1+ children, but also considering 0 vs 1, 0 vs 2 and 0 vs 2+. We only considered disease diagnoses that significantly increased childlessness from the main analyses. Overall, results were consistent, but the magnitude of the associations changed, especially when comparing 0 vs 1, where the effect was reduced compared to the main analyses (i.e. 0 vs 1+). We have added the following descriptions of these new major revisions and analyses to the manuscript:

We have included a new section in the **Results** called "Stratification by parity for individuals with children" (**Manuscript p 12-13**):

"In the main analyses we compared childless individuals to their siblings with children, regardless of parity. We reasoned that childless siblings might be more phenotypically similar to their siblings with fewer children (lower parity) compared to siblings who had multiple children (higher parity). When comparing childless individuals to their siblings with just one child, 14 disease diagnoses in women and

six in men had significantly reduced ORs (**Figure S4A-B**) compared to the main analysis. For example, in women, the OR of schizophrenia on childlessness dropped from 11.6 [9.1-14.9] to 5.1 [3.7-7.0] (P difference= 6.9×10^{-5}) when we compared childless individuals with their siblings with just one child. For individuals with higher parities, such as having exactly two children (**Figure SC-D**) or those with at least three or more children (**Figure S4E-F**), limited differences in ORs were observed compared to the main analyses. Jointly, these results indicated that childless individuals were more similar to their siblings with one child with regard to risk for childlessness-associated early-life diseases compared to siblings with higher parities.”

In the **Methods** section (**Manuscript p 27**), we added:

“For individuals with children, we further assessed whether the effects of disease diagnoses were consistent across parities by comparing childless individuals to their siblings with one child, two children, or more than two children.”

2. The authors also imply that the relationship between childlessness and major diseases will be consistent in other contexts, but the Nordic countries they use are really quite unusual. First, they have relatively high fertility in the European context, and a very strong 2-child norm, and those that have fewer children than 2 children are socially selected in various ways. In settings where childlessness is more common, and as childlessness becomes more normative, these associations with disease are likely to reduce. The relationship between fertility and various other health outcomes has been shown to be moderated by social context/social norms (underlined by the mediation by partnership status), which the authors don't acknowledge. Rather, the article is written as though this set of associations reflects a universal (biological) pattern, rather than being partly the product of a specific set of social and cultural conditions. I am very surprised to not see a richer discussion of childlessness and health, within broader cohort patterns, intergenerational processes of disadvantage, and social selection.

Answer: We thank the reviewer for the insightful comments and have added an extensive discussion about the generalizability of results to other cohorts and the impacts of non- biological factors on the observed associations.

First, we now have a more comprehensive section outlining the **context of fertility and childlessness in Finland and Sweden** in the **Supplementary Material p 6**:

“Childlessness is interesting to study in the Finnish and Swedish context. Despite declines in period fertility since around 2010, cohort fertility in Finland and Sweden has remained relatively stable and close to replacement levels^{1,2,3}. They have had higher levels of gender equity, rates of female labour force participation, and experienced a strong shift in women’s educational expansion⁴. Although fertility has remained relatively high, Finland has had the highest levels of childlessness in Europe⁵ and a ‘parity polarisation’ of high levels of childlessness coupled with higher parities in certain groups^{4,6}. There is also a parallel ‘partner polarisation’, or bifurcation of those who never partner with an increase in multi-partner fertility, given that a new union encourages childbearing^{3,7}.

Finland differs from Sweden in that it has a parity polarisation amongst particularly the lower, but also the medium-educated groups⁴. These differences have been attributed to Finland’s skewed national and local sex ratios related to deaths during WWII, mass emigration, low population density, and a larger proportion of women compared to men with tertiary education⁵. The ‘two-child family norm’ only seems to hold for the higher educated, and childlessness is higher amongst lower educated men and higher educated women⁴. Nordic gender equality and work-family reconciliation therefore seems to disproportionately aid the highly educated groups to reach the two-child norm. Conversely, the fertility patterns of lower educated men and women in Sweden and Finland are more heterogeneous and polarized between childless and higher parity⁴.”

Also, on **Supplementary Material p 38**:

“We also know that fertility transitions are conditioned on previous births and for that reason a parity-specific analysis is essential⁸. We therefore also look beyond the average fertility level of a country, since the average obscures the distribution of parity and polarization within particular subgroups. This is particularly important when examining childlessness in parity polarised countries⁹.“

Second, regarding the **generalizability of our results** to other countries, we have added to the discussion on **Manuscript p 19**:

“Finally, there is the question of the generalisability of our results to other nations. The Nordic countries have been the forerunners of demographic change in the realm of partnerships and fertility^{56,57}, making these results relevant for other industrialised nations. There are also important socioeconomic fertility gradients in Sweden and Finland that mirror changes in many industrialised nations. In the Nordic countries, there has been a positive association between men’s higher education and fertility^{58,59}. For Nordic women, there was initially the reverse gradient (i.e., higher levels of childlessness amongst those with higher education), but this has shifted over time with women increasingly similar to men⁵⁸. This ‘new Nordic’ fertility regime links higher socioeconomic status to higher fertility⁶⁰ but also multi-partner fertility⁶¹, a trend that has emerged in other nations as well.”

And on **Manuscript p 18**, we included the following discussion:

“Our findings are based on individuals born between 1956 and 1973 and results may thus not be entirely generalizable to more recent cohorts, because reproductive and partnering practices have changed and because better treatments might alleviate the effect of some diseases on childlessness.”

Third, to further address that our study is based on the context of Finland and Sweden, we have also **changed the title of the paper** as:

“The relationship of early-life disease with lifetime childlessness: Evidence from Finland and Sweden”.

Fourth, we have now included a more expanded section with appropriate citations highlighting the many **social, cultural, and structural factors contributing to childlessness, Manuscript p 3-4**.

*“Demographic research has isolated core factors linked to the rise in contemporary levels of childlessness. Access to effective contraception at the end of the 1960s in many countries and more recently, emergency post-coital contraception in the late 1990s, provided couples with more abilities to control their reproduction⁷. Women’s opportunities also vastly changed in many industrialised nations since the 1980s, with gains in higher levels of education and entry into the labour market accompanied with shifts in gender norms and equity^{11,12}. At the same time, couples faced work-life reconciliation constraints when planning to have children including lack of access to childcare, challenging housing conditions, economic uncertainty, inability to find a partner, and the general absence of supportive family policies¹². Given that women entered the labour market while still taking on the bulk of household labour, this ‘incomplete gender revolution’ resulted in women often being forced to decide between a career and parenthood^{13,14}. Higher levels of job strain and lack of work-life reconciliation has also been found to reduce fertility intentions¹⁵. Another shift has been the growth of precarious, temporary, and uncertain employment, leading many to postpone or forgo entering partnerships and parenthood^{16,17}. Many of these changes resulted in a general postponement of children to later ages¹². This in turn meant that individuals were having children at ages when they had lower fecundity (reproductive capacity), leading to infertility- related issues and having fewer children¹⁸. This also resulted in an increase in involuntary childlessness (see **Text Box**).*

Although economic, societal, and cultural freedom has been linked to individuals who remained

'childfree' or voluntarily childless¹⁹, the individuals who report that they never intended to have children, however, has remained low and is estimated to be around 5% in Europe³. Rather than being planned at an early age, becoming voluntarily childfree appears to be a mix of unforeseen circumstances, often related to postponement of childbearing and adaptation to a childfree life^{20,21}. This makes the decomposition of measuring voluntary versus involuntary childlessness challenging. A longitudinal study measuring fertility intentions of childless women (questioned from age 14 to their late 40s) found that they engaged in repeated postponement of childbearing and the subsequent adoption of a childless expectation at older ages or had indecision about parenthood evidenced by changing childless expectations across various ages."

Regarding the impacts of non-biological factors, we have also added to the discussion on **Manuscript p 16-17:**

"Our analyses highlight the important effects that many diseases have on partnership formation. Childless individuals are twice as likely to be single, in line with previous research showing the importance of selection into partnership on fertility²³. Singlehood represents a major mediator for the odds of being childless, especially in men, and we estimated that 29.3% (median) of the disease effect on childlessness in women and 37.9% in men was mediated by partnership formation. The effects of disease diagnoses on partnership formation are mediated by complex socio-cultural factors, social norms, and sex-specific behavioral preferences. For example, mental-behavioral disorders such as schizophrenia and acute alcohol intoxication were strongly associated with singlehood in men, but the effects were substantially smaller in women likely reflecting how different social and individual preferences for behavioral phenotypes correlated to mental-behavioral disorders impact partner choice in the two sexes."

Finally, we also added a new section and **Supplementary Table S1** that reviews existing evidence linking childlessness specifically to disease. The text in the **Main Manuscript p 5** is:

*"A review of the literature linking disease to childlessness can be found in **Table S1**. Previous medical studies largely only examined a limited selection of diseases directly associated with reproductive biology such as recurrent miscarriage²⁶, polycystic ovary syndrome²⁷, and endometriosis²⁸. Another large area of research has been the study of infertility and inflammatory bowel disease (IBD). A systematic review found that up to one-third of IBD patients opted to be voluntarily childless, often related to lack of knowledge related to pregnancy-related IBD issues²⁹, with clinical studies finding that women with moderate to severe IBD had higher incidences of adverse pregnancy outcomes³⁰. Multiple sclerosis has also been associated with higher levels of childlessness, linked to fewer being in a stable relationship, fear of genetically transmitting multiple sclerosis, or discontinuation of treatments³¹. Another line of study has examined fecundity in patients with major psychiatric disorders³², and those with a genetic liability for schizophrenia, with the later finding no clear association with childlessness³³."*

3. The last concern is that the analysis is not accurately represented in the title, abstract and interpretation. The main analysis - the sibling design used for the main analysis is very robust, in that it cuts the observation time at one year prior to first childbirth in the parous sibling. But what this means is that the study focuses on diseases occurring very early in the life course and are severe enough to require hospitalisation. This gives a very different perspective to considering all major diseases. Therefore, although from a scientific point of view, the analysis is impressive, I think the framing/interpretation should be further nuanced, to avoid contributing to social stigma experienced by childless individuals, and to represent the evidence the study provides more accurately.

Answer: The reviewer raises an important point and we have now gone through the entire paper and rewritten the framing and interpretation to make it more nuanced and avoid any stigmatising language about childless individuals. We have also added in a new **Text Box** which defines childlessness and unpacks the various debates in this area.

The reviewer also notes that we look at diseases early in the life course that are severe enough to cause hospitalisation, which we agree provides a different perspective than all major diseases. An advantage of our approach is that we are able to include diseases prior to childbearing. In our final discussion, we have reflected on the strengths and weaknesses of our measures and noted the points raised by the reviewer.

We have changed the title to *“The relationship of early-life diseases with lifetime childlessness: Evidence from Finland and Sweden”*.

Detailed comments:

Abstract:

4. The first sentence of the abstract is extremely vague. What countries in the world are these percentages based on? Globally? High income countries? Is childlessness increasing everywhere?

Answer: We thank the reviewer for the comment. It now reads:

“The percentage of women born 1965-1975 remaining childless is ~20% in many Western European and ~30% in some East Asian countries.”

The full references for these sources are listed in the first lines of the manuscript.

5. I think the abstract misses an explanation of what we already know, which is considerable. There have been many studies based a) cause of death data b) prescription data/disease registers c) survey data already. We already know that nulliparous individuals are more likely to have certain types of health behaviours and health conditions. What is missing - and this study provides- is a more comprehensive picture going into detail of these diseases prior to childbearing.

Answer: First, we have now **added the following lines to the abstract:**

“Single diseases have been linked to childlessness, mostly in women, yet we lack a comprehensive picture of the effect of early-life diseases on lifetime childlessness.”

Second, we have **added a new Supplementary Table S1 and a new section in the paper** reviewing previous studies linking childlessness to health behaviour and conditions (see text in our above response to question 2).

Introduction:

6. I find the introduction presents the issues a bit simplistically and does little to place childlessness in social and cultural contexts.

Answer: The reviewer is correct and when cutting content in the previous draft of the paper we removed too much detail on the rich demographic and socioeconomic literature regarding childlessness. We have included a more expanded section with appropriate citations, **Manuscript p 3-4**, which we included in our above response to question 2.

7. Again, the first sentence needs some context- I see you reference Sobotka, but which countries exactly, and what time period- needs to be more specific. The second sentence uses data on nulliparity at age 40- some of this is related to postponement.

Answer: We thank the reviewer for the comment and have revised the sentence as follows (**Manuscript p 3**):

“In Western European countries, around 20% of women born around 1965 remained childless, with the highest levels of permanent childlessness now in East Asia, ranging from 28% (Japan) to 35% (Hong Kong) for women born in 1975^{7,8}.”

8. I really object to the painting of childlessness as a problem that needs to be addressed- why does it present a challenge for societal sustainability? Some people would really argue the opposite (not that I support such polarised perspectives, either). I think this is done to develop the motivation for the paper but this could be done in a more nuanced way.

Answer: The reviewer is completely correct and this was inadvertent and has been changed throughout. We have also added a new **Text Box**, which clarifies the definition and various positions, which as you say can be quite polarised.

On **Manuscript p. 3**.

“Text box: Defining Childlessness

We use the term childlessness to describe individuals that have had no live-born children by the end of their reproductive lifespan (age 45 for women; 50 for men). Childlessness is defined in the literature as being both involuntary, related to biology and fecundity (e.g., infertility, inability to find a partner) and voluntary or ‘childfree’¹ (e.g., active choice, preference²). It has been estimated that 4-5% of the current 15-20% women who are childless in Europe are voluntary childless³. Childless individuals are subjected to discrimination and marginalization in many societies⁴, with infertile women globally experiencing multiple types of violence and coercion⁵. A parallel line of work, which is not the position of this paper or authors, is to problematize and stigmatize childless individuals as egoistic and place blame on this group for producing a so-called ‘demographic disaster’ of shrinking and ageing populations and collapse of social security systems⁶. The approach of this paper is to provide a neutral, data-driven, and factual examination of early-life diseases related to childlessness, with the aim to design a better understanding of health to prevent childlessness among those who want to have children.”

9. I am very happy to see a conceptual framework which describes the possible mediators between disease and childless status, but again it's a bit simplistic. What is the role of intergenerational health/fertility preferences, for example? Could it be the case that challenges conceiving could affect health behaviours and then disease status, in turn? Then, for the comparison group, who have children, depending on the number of timing, this affects their health behaviours and disease profiles, so it makes the childless individuals seems more/less likely to suffer from some conditions.

Answer: We thank the reviewer for the suggestion and have revised **Figure S1 (Supplementary Materials p 2)** accordingly.

Figure S1: Theoretical framework for the relationship of diseases with childlessness.

First, we have added “*Intergenerational health: fear of transmitting disease to newborn*” as one of the possible mediators for disease-childlessness associations and add “*Fertility preferences and intentions*”. We included fertility preferences and intentions, since the literature in this area covers broader preference formation around intergenerational transmission alone.

Second, we have added a dashed arrow to indicate that there could be reverse causation between involuntary childlessness and disease status, since involuntary childlessness might influence an individual’s health behaviors and likewise disease status. However, by applying a matched sibling design to rich longitudinal register data, our study aimed to estimate the relationship between early-life diseases and lifetime childlessness while avoiding the impact of possible reverse causation.

Accordingly, we have added the following descriptions to the **Supplementary Materials p 2**:

“There could also be concerns about intergenerational transmission of disease and the intergenerational transmission of health and fertility preferences.”

“While individuals’ health conditions can also be influenced by their childless status, our study focused on the path from early-life diseases to lifetime chances of being childless.”

10. The third paragraph- if it's intended to be a literature review of what is already known, ignores existing (more demographically) oriented studies which use cause of death data, survey data etc, which already point to the relationship between parity and various diseases (CVD, cancer, alcohol use etc).

Answer: This is an excellent point and as outlined previously, we have now completely rewritten the introduction to include more depth and a review of the demographic and disease related literature in

this area. we also added a new section and **Supplementary Table S1** that reviews existing evidence linking childlessness specifically to disease. However, we note that engaging in a deeper systematic review of the literature covering the relationship between parity and various diseases goes beyond the scope of the current article and available space.

Methods:

11. The Figure is a very neat way of describing the analysis sample. In the text, it is not clear to me why you include individuals across three generations. I think you could include mention of how you select siblings within same-sex sibling groups of more than 2. I also think the fact that you drop 80% of the sample should be reflected on- how are these individuals different?

Answer: We thank the reviewer for the excellent suggestion, which we have now followed and now explained the reason why we included individuals across three generations and also described how we selected same-sex sibling pairs among families with more than 2 siblings:

“Panel A shows the birth cohorts, the follow-up period, and the number of Finns and Swedes included in the analyses. A total of three generations of familial relationships were considered for the index persons, with the parental information used to match sibling pairs, the children’s information to define the main outcome of childlessness, and the registered spouse’s information to determine the secondary outcome of singlehood. Panel B shows the statistical approach used in the main analyses, where only families with same-sex siblings discordant for childlessness are included. Within each family, we randomly selected one childless sibling as a case, and, as control, the sibling with children that was the closest for birth order with the case. Within a sibling pair, disease diagnoses are only considered if they occurred at least one year before the birth of the first child or within the corresponding age at which the sibling was childless, since sibling ages will differ.”

Additionally, for **Figure 1a (Manuscript p 6)**, we have changed “N of same-sex full-sibling pairs” to “N of families with same-sex full siblings”.

12. One aspect which I think deserves better representation in the paper. So, if in the sibling analysis, you cut the observation time one year before the parous sibling has children, then what is the mean age of cutting the time? This basically means that you are observing diseases that occur only in really young people. So, the statements later on about observing childless individuals over their reproductive lifespan is only true for the robustness checks. The main analysis - which most of the paper is based on - is on diseases that occur mainly before aged 30? If so, I think the title, abstract and interpretation of data need to adjusted to make this clearer.

Answer: We agree with the reviewer that in order to avoid reverse causation we only considered diseases diagnosed one year prior to the parous sibling having children, with the average age of cutting the time corresponding to 28.2 (SD=5.6) in Finnish men, 28.4 (5.7) in Swedish men, 26.4 (5.4) in Finnish women, and 26.5 (5.4) in Swedish women. However, as shown in our age onset analyses, there are also sibling pairs who had children relatively late in their lifespan. This is especially true for men. Taking the Swedish population as an example, 37.1% of male sibling pairs and 27.8% of female sibling pairs had their first children (cutting time) after age 30.

To further address this, we have added the following descriptions to **Methods (Manuscript p 27)**:

“With the matched design, we excluded diseases diagnosed after the reproductive lifespan in order to avoid reverse causation.”

We have also changed the **title** of the paper to reflect this suggestion:

“The relationship of early-life disease with lifetime childlessness: Evidence from Finland and Sweden”

13. One factor which is not mentioned. You combine the data for these two countries: what's the justification for using both? why bother, especially when the disease data are sometimes a little different? Linked to the above, is this a sample size issue? Given that these are quite young people, are some of these diseases quite rare?

Answer: This is an excellent point, and we now provide more depth about why it is interesting to study these two countries and added the substantive text to the **Supplementary Material p 6**:

“Childlessness is interesting to study in the Finnish and Swedish context. Despite declines in period fertility since around 2010, cohort fertility in Finland and Sweden has remained relatively stable and close to replacement levels^{1,2,3}. They have had higher levels of gender equity, rates of female labour force participation, and experienced a strong shift in women’s educational expansion⁴. Although fertility has remained relatively high, Finland has had the highest levels of childlessness in Europe⁵ and a ‘parity polarisation’ of high levels of childlessness coupled with higher parities in certain groups^{4,6}. There is also a parallel ‘partner polarisation’, or bifurcation of those who never partner with an increase in multi-partner fertility, given that a new union encourages childbearing^{3,7}.

Finland differs from Sweden in that it has a parity polarisation amongst particularly the lower, but also the medium-educated groups⁴. These differences have been attributed to Finland’s skewed national and local sex ratios related to deaths during WWII, mass emigration, low population density, and a larger proportion of women compared to men with tertiary education⁵. The ‘two-child family norm’ only seems to hold for the higher educated, and childlessness is higher amongst lower educated men and higher educated women⁴. Nordic gender equality and work-family reconciliation therefore seems to disproportionately aid the highly educated groups to reach the two-child norm. Conversely, the fertility patterns of lower educated men and women in Sweden and Finland are more heterogeneous and polarized between childless and higher parity⁴.”

In the discussion on **Manuscript p 19** we also write:

“Finally, there is the question of the generalisability of our results to other nations. The Nordic countries have been the forerunners of demographic change in the realm of partnerships and fertility^{56,57}, making these results relevant for other industrialised nations. There are also important socioeconomic fertility gradients in Sweden and Finland that mirror changes in many industrialised nations. In the Nordic countries, there has been a positive association between men’s higher education and fertility^{58,59}. For Nordic women, there was initially the reverse gradient (i.e., higher levels of childlessness amongst those with higher education), but this has shifted over time with women increasingly similar to men⁵⁸. This ‘new Nordic’ fertility regime links higher socioeconomic status to higher fertility⁶⁰ but also multi-partner fertility⁶¹, a trend that has emerged in other nations as well.”

We also combined data from two countries for two main reasons. First, we wanted to test whether the observed associations are consistent between the two countries, which would increase the robustness of the results. If the effects of the same disease on childlessness are heterogeneous across two populations, this might indicate differences in social and cultural contexts and/or healthcare systems. However, remarkably, we observed very robust effects in the two countries. Second, we meta-analyzed results from two countries to improve the statistical power of the analyses.

To clarify this, we have rephrased the following sentence in **Discussion (Manuscript p 17)**:

“The use of nationwide data from two countries provided a large sample of 2.5 million, and most importantly, allowed us to assess how robust our findings were in different healthcare systems and diagnostic practices.”

14. Marriage and partnership. How does the data account for cohabitating unions? I am not sure but I think for at least some of that period in Sweden data are available only for marriages.

Answer: We thank the reviewer for the comment. To clarify this concept, we have rephrased the following sentences in **Methods (Manuscript p 25)**:

“For index individuals in Finland, we obtained their longitudinal marriage and partnership information between 11 April 1971 and 31 December 2018 from the Population Information System. In Sweden, we collected information of married couples and cohabiting unions with biological children between 1 January 1977 and 31 December 2017 from Statistics Sweden. We defined individuals as partnerless if they did not have any abovementioned marriage or partnership registered by age 45 (women) or 50 (men).”

15. Disease data. If the cancer registry is only available for Finland, how do they Swedish and Finnish cancer rates compare? Even if disease prevalence for cancer is similar, I would assume that Finns are 'diagnosed' at an early stage, because Swedes get their cancer data from hospitalisation? Or does it not vary? If it does, I would be tempted to keep the disease data identification identical, or to justify better why its ok to keep it different.

Answer: We thank the reviewer for the comment. Overall, cancer diagnoses are well captured by the in-patient/outpatient registers that are available in both Finland and Sweden. The advantage of including the cancer registers is that, in Finland, its coverage dates back as early as 1953. There were only two cancer diagnoses in both Finland and Sweden (**Table S5**), including lymphoid leukemia and myeloid leukemia. For these two leukemia diagnoses, as mentioned in **Table S6**, we excluded them from further analyses since receiving diagnoses of these diseases resulted in an almost complete lack of children in the observational period. Additionally, benefiting from the earliest coverage of the Cancer Register in Finland, we were able to include three additional cancer diagnoses available only in Finland, including malignant neoplasm of testis, malignant neoplasm of brain, and secondary and unspecified malignant neoplasm of lymph nodes.

16. Design. I think you overstate the ability of sibling FE to control for confounding. Siblings are not that similar. Siblings share 50% genetic material, similar family of origin characteristics, but can experience quite different childhood conditions. For example, birth order and family size for the majority of their time can result in quite different health profiles. As you adjust for year of birth, this takes account of who is older and by how much, but that's it, really. What about the other siblings in the family? Parental status - marriage- divorced and the age at which this occurred?

Answer: We agree with the reviewer. Following your suggestions, we have extended the limitations of our study in the **Discussion** section (**Manuscript p 18-19**):

“Fourth, despite our attempt to use a study design that minimizes confounding and reverse causality, these biases might challenge the interpretation of the results. For example, although full siblings share roughly half of their genetic materials and similar family of origin characteristics and only siblings with the closest birth order were considered, siblings can still experience different childhood conditions and even be influenced by each other's conditions⁵⁵.”

Results:

17. Given the conceptual framework you include, I think it would make sense to have a more detailed set of descriptives which describe the social characteristics of the childless/controls in the two countries. I don't see anything which describes the education levels of the parents, health patterns during childhood, partnership patterns, parity, social class, education, of the cases and controls.

Answer: We have added a new section called “Descriptive statistics of fertility and social characteristics in Finland and Sweden” to the **Supplementary Materials p 6-9** to describe childlessness and parity (**Table S2**) as well as social characteristics of the childless cases and controls (**Table S3-4**) for both sexes in the two countries. Regarding social characteristics, we provided statistics on parental education level and index person’s education level for both countries and partnership patterns for Finland, depending on data availability.

Table S2: Descriptive statistics of childlessness and parity for index persons in Finland and Sweden.

	Finland		Sweden	
	Women	Men	Women	Men
Childless	107,913 (18.8 %)	128,213 (27.7 %)	122,285 (14.3%)	151,241 (23.1 %)
One child	91,114 (15.9 %)	69,047 (14.9 %)	115,262 (13.5%)	91,426 (13.9 %)
Two children	206,436 (36.1 %)	145,292 (31.4 %)	377,564 (44.3%)	250,122 (38.1 %)
Three children	114,966 (20.1 %)	81,764 (17.6 %)	177,225 (20.8%)	118,518 (18.1 %)
At least four children	52,089 (9.1 %)	39,094 (8.4 %)	60,786 (7.1%)	44,663 (6.8 %)
Total	572,518 (100%)	463,410 (100%)	853,122 (100%)	655,970 (100%)

Table S3: Descriptive statistics of social characteristics for index persons in Finland.

Variable	Women		Men	
	With children	Childless	With children	Childless
Parental education level (ISCED 1997)				
Pre-primary education, primary education or first stage of basic education, or lower secondary or second stage of basic education (Level 0-2)	224,817 (48.4%)	48,904 (45.3%)	177,274 (52.9%)	72,107 (56.2%)
Upper secondary education (Level 3)	141,466 (30.4%)	32,438 (30.1%)	92,034 (27.5%)	33,818 (26.4%)
Post-secondary non-tertiary education (Level 4)	399 (0.1%)	90 (0.1%)	95 (0%)	24 (0%)
First stage of tertiary education (Level 5)	95,139 (20.5%)	25,710 (23.8%)	64,076 (19.1%)	21,735 (17.0%)
Second stage of tertiary education (Level 6)	2,784 (0.6%)	771 (0.7%)	1,718 (0.5%)	529 (0.4%)
Index person's education level (ISCED 1997)				
Pre-primary education, primary education or first stage of basic education, or lower secondary or second stage of basic education (Level 0-2)	41,466 (8.9%)	13,232 (12.3%)	52,867 (15.8%)	31,625 (24.7%)
Upper secondary education (Level 3)	184,774 (39.8%)	38,776 (35.9%)	157,338 (46.9%)	64,247 (50.1%)
Post-secondary non-tertiary education (Level 4)	7,992 (1.7%)	1,170 (1.1%)	5,800 (1.7%)	965 (0.8%)
First stage of tertiary education (Level 5)	224,983 (48.4%)	53,117 (49.2%)	115,052 (34.3%)	30,419 (23.7%)
Second stage of tertiary education (Level 6)	5,390 (1.2%)	1,618 (1.5%)	4,140 (1.2%)	957 (0.7%)
Partnership pattern				
No partner	69,263 (14.9%)	70,300 (65.1%)	45,308 (13.5%)	93,067 (72.6%)
Ever-married	395,127 (85.0%)	37,169 (34.4)	289,834 (86.5%)	34,719 (27.1%)
Registered partnership	215 (0.05%)	444 (0.41%)	55 (0.02%)	427 (0.33%)

Table S4: Descriptive statistics of social characteristics for index persons in Sweden.

Variable	Women		Men	
	With children	Childless	With children	Childless
Parental education level (ISCED 1997)				
Pre-primary education, primary education or first stage of basic education, or lower secondary or second stage of basic education (Level 0-2)	227,829 (31.2%)	38,260 (31.3%)	179,183 (35.5%)	58,593 (38.7%)
Upper secondary education (Level 3)	331,431 (45.3%)	52,960 (43.3%)	217,967 (43.2%)	62,713 (41.5%)

Post-secondary non-tertiary education (Level 4)	13,745 (1.9%)	2,460 (2.0%)	7,705 (1.5%)	2,131 (1.4%)
First stage of tertiary education (Level 5)	149,146 (20.4%)	26,934 (22.0%)	94,453 (18.7%)	26,311 (17.4%)
Second stage of tertiary education (Level 6)	8,686 (1.2%)	1,671 (1.4%)	5,421 (1.1%)	1,493 (1.0%)
Index person's education level (ISCED 1997)				
Pre-primary education, primary education or first stage of basic education, or lower secondary or second stage of basic education (Level 0-2)	53,253 (7.3%)	14,833 (12.1%)	71,808 (14.2%)	31,105 (20.6%)
Upper secondary education (Level 3)	362,598 (49.6%)	55,670 (45.5%)	274,402 (54.4%)	80,326 (53.1%)
Post-secondary non-tertiary education (Level 4)	39,626 (5.4%)	7,938 (6.5%)	49,607 (9.8%)	13,314 (8.8%)
First stage of tertiary education (Level 5)	267,718 (36.6%)	42,304 (34.6%)	101,521 (20.1%)	24,897 (16.5%)
Second stage of tertiary education (Level 6)	7,642 (1.0%)	1,540 (1.3%)	7,391 (1.5%)	1,599 (1.1%)

Accordingly, we included the following descriptions in the **Manuscript p 7**:

“The proportion of childless individuals – those with no live-born children – was higher in men (25.4%) than women (16.6%), higher amongst Finns (23.3%) than Swedes (18.7%), and slightly higher in the general population than in individuals who had same-sex full-siblings (0.9% and 0.5% higher in women and men, respectively). Among individual with children, a two-child parity was most prevalent in men (47.1%) and women (48.9%), amongst both Finns (44.0%) and Swedes (50.8%) (Table S2). Both women and men with the lowest educational attainment were more likely to be childless (e.g., in Finland, 24.2% were childless in women and 37.4% in men) than the general population (e.g., in Finland, 18.8% were childless in women and 27.7% in men) (Table S3-S4). When looking across generations, the education level of parents was correlated with an individual's childless status (Table S3-S4). For example, among index persons whose parents had completed the first stage of tertiary education, men were less likely to be childless (e.g., in Finland, 25.3%) compared to the general population (e.g., in Finland, 27.7%) while women behaved in the opposite way (e.g., in Finland, 21.3%, compared to 18.8% in the general population).”

Additionally, we added the following details to the **Supplementary Materials p 6-7**: For childlessness and parity, we added:

“For index individuals in both Finland and Sweden, we considered the numbers of biological children they had by age 45 in women and 50 in men (Table S2). Among 1,425,640 index women and 1,119,380 men, 230,198 women and 279,454 men were childless. The proportion of childless individuals – those with no live-born children – was higher in men (25.4%) than women (16.6%), and higher amongst Finns (23.3%) than Swedes (18.7%). Among index persons with children, a two-child parity was most prevalent in men (47.1%) and women (48.9%), amongst both Finns (44.0%) and Swedes (50.8%).”

For social characteristics of the childless cases and controls, we added:

“To understand the extent to which the social characteristics varied amongst childless individuals and individuals with children in the general population, we considered the highest education level and parental education level in both Finns and Swedes and partnership patterns in Finns only (Table S3-S4),

depending on data availability.

Overall, in both countries, women were more educated than men (e.g., in Finland, 49.8% of index women completed the first stage of tertiary education while the proportion dropped to 32.5% for index men). Both women and men with the lowest educational attainment (level 0-2; pre-primary education, primary education or first stage of basic education, or lower secondary or second stage of basic education) were more likely to be childless (e.g., in Finland, 24.2% were childless in women and 37.4% in men) than the general population (e.g., in Finland, 18.8% were childless in women and 27.7% in men). Amongst individuals who completed the first stage of tertiary education (levels 5 and 6), men were less likely to be childless (e.g., in Finland, 20.8%) compared to the general population (e.g., in Finland, 27.7%), however, for women, a slightly higher probability of childlessness was observed (e.g., in Finland, 19.2%, compared to 18.8% in the general population).

Across generations, individuals included in these analyses were more educated than their parents (e.g., in Finland, 42.1% of index persons completed the first stage of tertiary education while the proportion dropped to 20.5% in their parents). The education level of parents was also correlated with an individual's childless status. For example, men who had parents with the lowest educational attainment (level 0-2) were more likely to be childless (e.g., in Finland, 28.9%) than the general population (e.g., in Finland, 27.7%). Among index persons whose parents had completed the first stage of tertiary education (levels 5-6), men were less likely to be childless (e.g., in Finland, 25.3%) compared to the general population (e.g., in Finland, 27.7%) while women behaved in the opposite way (e.g., in Finland, 21.3%, compared to 18.8% in the general population).

18. Fig 2: You need to include CIs on the main left hand side graphs. Some of the ORs for individual conditions are enormous but I'm assuming they also have very large CIs, as they are quite rare. It's also not very clear what the labels on the charts (left hand side) refer to, I think you need some lines to link the labels to the correct marker.

Answer: We thank the reviewer for the comment and have revised the labels of **Figure 2 (Manuscript p 8)** accordingly. For CIs, we think the figure will become too busy by including CIs of >400 disease endpoints, and ignoring CIs is also a common way of showing results in many phenome-wide association studies (PheWAS). However, the main panel of our interactive online dashboard (<https://dsgelrs.shinyapps.io/DiseaseSpecificLRS/>) shows the CIs of every selected endpoint.

19. I find that the analysis of # children unborn etc, although allows some interesting statements to be made, is not very useful, because it assumes that the disease led to the childlessness, rather than be the other way around or be mutually reinforcing.

Answer: Our aim in this section was to try to produce quantifiable public health policy. This has been done previously in fertility research such as the calculation of the contribution of assisted reproduction to completed fertility (Sobotka et al. 2008, PDR). We recognise that causality is an issue, but since we cannot do a RCT, it is not a problem that can be easily solved. The sibling design that we adopted was the best solution possible and remains conservative given that siblings also partially share genetics. To nuance this, we have also now adapted the text **Manuscript p 13** and added:

"We note that the estimates of the number of children not born or 'missing' due to disease diagnoses could be slightly biased because of the use of only individuals with same-sex full siblings in our main analyses."

20. I would appreciate in the discussion, some more nuanced reflection on unique context (as discussed above), beyond the generic statement that it should be replicated in other populations. Further, as voluntary childlessness becomes a more important component in future, these results might lose their relevance. The limitations could be more detailed. The study presents this data, which does not include primary care data, to be representative of the onset and occurrence of major diseases. Some diseases are very unlikely to be well covered and the bias will be for capturing more severe aspects of mental health and behavioural conditions. Others such as low grade metabolic conditions may never be captured using these data sources.

Answer: We thank the reviewer for the insightful comments and have improved and rewritten the discussion accordingly.

Regarding the generalizability of results to other countries (**Manuscript p 19**):

“Finally, there is the question of the generalisability of our results to other nations. The Nordic countries have been the forerunners of demographic change in the realm of partnerships and fertility^{56,57}, making these results relevant for other industrialised nations. There are also important socioeconomic fertility gradients in Sweden and Finland that mirror changes in many industrialised nations. In the Nordic countries, there has been a positive association between men’s higher education and fertility^{58,59}. For Nordic women, there was initially the reverse gradient (i.e., higher levels of childlessness amongst those with higher education), but this has shifted over time with women increasingly similar to men⁵⁸. This ‘new Nordic’ fertility regime links higher socioeconomic status to higher fertility⁶⁰ but also multi-partner fertility⁶¹, a trend that has emerged in other nations as well.”

And on **Manuscript p 18**, we included the following discussion:

“Our findings are based on individuals born between 1956 and 1973 and results may thus not be entirely generalizable to more recent cohorts, because reproductive and partnering practices have changed and because better treatments might alleviate the effect of some diseases on childlessness.”

Regarding the importance of voluntary childlessness, we have substantially improved the discussion on this topic by provided a detailed discussion in the introduction (**Manuscript p 4**) and in the limitation of the discussion (**Manuscript p 18**):

“Second, as with previous studies to date, we were unable to partition the impacts that diseases have into voluntary and involuntary childlessness due to a lack of data on reproductive preferences and intentions and difficulties in differentiating the two^{22,52,53}. Given that estimates of the voluntary childfree group are around a quarter of total childless, we note that public health interventions should take this group of voluntary childfree individuals into account.”

Regarding the coverage of disease diagnoses (**Manuscript p 18**):

“First, disease diagnoses are largely obtained from registers covering secondary and tertiary healthcare in hospitals that capture more severe disease cases (e.g., mental health and behavioral conditions), and thus, lack of the onset and occurrence of major diseases included in primary healthcare data (e.g., low-grade metabolic conditions).”

21. The final statement- about the public health impact- could be further developed, based on the acknowledgement that the study focuses on early-life diseases.

Answer: We thank the reviewer for the suggestion. The final statement has been revised accordingly

(Manuscript p 19):

“In conclusion, we comprehensively described the effects different diseases, particularly those with onset prior to the peak reproductive age have on the chance of being childless over a lifetime. This evidence can be used as a basis for future studies focusing on prioritizing health interventions to counter involuntary childlessness.”

Reviewer #2:

This manuscript sought to assess the relationship of disease to childlessness, across two national (Scandinavian) cohorts. Strengths of the study include the population-level data, the breadth of diseases considered (and the attendant methods used to reign in their numbers for informative results), and the care taken to assess associations by gender.

My primary concerns with the manuscript are as follows:

22. I wonder about the choice of the cohort (birth age 1956-1973); it seems ideal to include the widest cohort possible, particularly given changes in diagnostic processes over time (e.g., might diagnoses be coming earlier, generally, for instance?). Individuals born into the 80s are at or near 40 at present, and would offer an expanded group of individuals with more current medical histories who have passed through a substantial portion of child-bearing/rearing years. This would seem particularly important for the mental/behavioral disorders found to be central drivers in the analyses? Related: are there cohort effects that could be examined here (beyond sensitivity analyses) controlling for age at diagnosis; as our understanding of the burden of disease has advanced, it stands to reason changes have occurred in the relationship of disease to childlessness?

Answer: We thank the reviewer for the very insightful comments. The main reason to define index persons as women born in 1956-1973 and men in 1956-1968 was to largely cover complete follow-up until the end of the reproductive period (age 45 in women and 50 in men). We think, as one of the strengths of our study, the design of embracing virtually complete coverage of health and reproductive information advanced our understanding of the impact different diseases had on the chance of being childless over a lifetime.

We agree with the reviewer that by including more recent birth cohorts we can further examine the extent to which the identified disease-childlessness associations from the old cohort stayed with advances in healthcare. However, we currently don't have data access to the more recent birth cohorts. We agree that access to more recent cohorts will provide better coverage for certain diseases. We have now added in different parts of the manuscript that our focus is on early-life diseases.

We have changed the title to:

“The relationship of early-life diseases with lifetime childlessness: Evidence from Finland and Sweden”.

We included the following context in the discussion section, as one of the limitations of our study **(Manuscript p 18):**

“Our findings are based on individuals born between 1956 and 1973 and results may thus not be entirely generalizable to more recent cohorts, because reproductive and partnering practices have changed and because better treatments might alleviate the effect of some diseases on childlessness.”

We specified in the conclusion that our study focus on diseases onset prior to the peak reproductive age **(Manuscript p 19):**

“In conclusion, we comprehensively described the effects different diseases, particularly those with onset prior to the peak reproductive age have on the chance of being childless over a lifetime. This evidence can be used as a basis for future studies focusing on prioritizing health interventions to counter involuntary childlessness.”

We however notice that, for mental-behavioral diseases, the improvement in medical care has been limited over the past decades.

23. The manuscript makes clear its intention to provide evidence which 'can be used for targeted health interventions to counter decreasing fertility, reproductive health, involuntary childlessness, and shrinking populations.' However, the methods do not offer any means of assessing whether - and to what degree - the childlessness of its disease-affected populations are attributable to meaningfully targetable reasons (indeed, characterizing even 'voluntary' reasons as changeable dismisses the economic and personal realities that shape the 'choice'). This is a non-trivial issue, as there are many diseases for which pregnancy is counter-indicated (whether due to dangers posed to the mother/child-bearer, or risk for offspring health), which pose direct, biological challenges to reproduction, or whose costs limit the feasibility of child-rearing.

Answer: We have now addressed this issue in considerable detail when we **extensively rewrote the introduction** to include:

First, a more detailed definition of childlessness, including voluntary versus involuntary in a new **Text Box (Manuscript p 3)**:

“Text box: Defining Childlessness

We use the term childlessness to describe individuals that have had no live-born children by the end of their reproductive lifespan (age 45 for women; 50 for men). Childlessness is defined in the literature as being both involuntary, related to biology and fecundity (e.g., infertility, inability to find a partner) and voluntary or ‘childfree’¹ (e.g., active choice, preference²). It has been estimated that 4-5% of the current 15-20% women who are childless in Europe are voluntary childless³. Childless individuals are subjected to discrimination and marginalization in many societies⁴, with infertile women globally experiencing multiple types of violence and coercion⁵. A parallel line of work, which is not the position of this paper or authors, is to problematize and stigmatize childless individuals as egoistic and place blame on this group for producing a so-called ‘demographic disaster’ of shrinking and ageing populations and collapse of social security systems⁶. The approach of this paper is to provide a neutral, data-driven, and factual examination of early-life diseases related to childlessness, with the aim to design a better understanding of health to prevent childlessness among those who want to have children.”

Second, a new section outlining the demographic and socioeconomic reasons associated with childlessness (**Manuscript p 3-4**):

“Demographic research has isolated core factors linked to the rise in contemporary levels of childlessness. Access to effective contraception at the end of the 1960s in many countries and more recently, emergency post-coital contraception in the late 1990s, provided couples with more abilities to control their reproduction⁷. Women’s opportunities also vastly changed in many industrialised nations since the 1980s, with gains in higher levels of education and entry into the labour market accompanied with shifts in gender norms and equity^{11,12}. At the same time, couples faced work-life reconciliation constraints when planning to have children including lack of access to childcare, challenging housing

conditions, economic uncertainty, inability to find a partner, and the general absence of supportive family policies¹². Given that women entered the labour market while still taking on the bulk of household labour, this ‘incomplete gender revolution’ resulted in women often being forced to decide between a career and parenthood^{13,14}. Higher levels of job strain and lack of work-life reconciliation has also been found to reduce fertility intentions¹⁵. Another shift has been the growth of precarious, temporary, and uncertain employment, leading many to postpone or forgo entering partnerships and parenthood^{16,17}. Many of these changes resulted in a general postponement of children to later ages¹². This in turn meant that individuals were having children at ages when they had lower fecundity (reproductive capacity), leading to infertility-related issues and having fewer children¹⁸. This also resulted in an increase in involuntary childlessness (see **Text Box**).”

Third, a new section summarizing previous literature linking disease to childlessness where we explicitly address voluntary childlessness due to concerns for offspring health (**Manuscript p 5**):

“A review of the literature linking disease to childlessness can be found in **Table S1**. Previous medical studies largely only examined a limited selection of diseases directly associated with reproductive biology such as recurrent miscarriage²⁶, polycystic ovary syndrome²⁷, and endometriosis²⁸. Another large area of research has been the study of infertility and inflammatory bowel disease (IBD). A systematic review found that up to one-third of IBD patients opted to be voluntarily childlessness, often related to lack of knowledge related to pregnancy-related IBD issues²⁹, with clinical studies finding that women with moderate to severe IBD had higher incidences of adverse pregnancy outcomes³⁰. Multiple sclerosis has also been associated with higher levels of childlessness, linked to fewer being in a stable relationship, fear of genetically transmitting multiple sclerosis, or discontinuation of treatments³¹. Another line of study has examined fecundity in patients with major psychiatric disorders³², and those with a genetic liability for schizophrenia, with the later finding no clear association with childlessness³³.”

Fourth, we adapted **Figure S1** to include the multiple aspects contributing to childlessness and it’s relationship to disease.

“Figure S1: Theoretical framework for the relationship of diseases with childlessness.”

Fifth, given that linking childlessness to shrinking populations can be stigmatising, also recognised by another reviewer, we have also removed that statement.

In the discussion of limitations, we also added **(Manuscript p 21-22)**:

“Second, as with previous studies to date, we were unable to partition the impacts that diseases have into voluntary and involuntary childlessness due to a lack of data on reproductive preferences and intentions and difficulties in differentiating the two^{22,52,53}. Given that estimates of the voluntary childfree group are around a quarter of total childless, we note that public health interventions should take this group of voluntary childfree individuals into account”

24. The authors acknowledge this, to an extent, in stating that 'there are multiple non-disease related reasons for individuals to postpone parenthood or not have children ... individual choice, work-life reconciliation, housing and economic uncertainty, or inability find a partner.' However, no attempt seems to be made to integrate the majority of these features into the analysis or, at any length/depth, in their discussion. Certainly, data relevant to these features are available in the Scandinavian registers (e.g., education, unemployment histories?) - are the authors able to integrate? Given the long (and, indeed, lengthening) timespan constituting the 'child bearing/rearing' years, use of same-sex siblings (who will only share early-life exposures) does not address the likely differential environments characterizing the affected vs. unaffected siblings through their reproductive adulthoods. Likewise, such information would be a valuable covariate in the sensitivity analyses abandoning the sibling

structure.

Answer: We thank the reviewer for this suggestion. We have collected education levels for both index person and their parents from nationwide registers of both countries, but we currently don't have data access to individuals' unemployment histories. Therefore, our response to this comment will be focused on educational attainment only.

First, we included a detailed description of how educational attainment is related to childlessness and fertility in the Finnish and Swedish context on **Manuscript p 8**.

"Finland differs from Sweden in that it has a parity polarisation amongst particularly the lower, but also the medium-educated groups (Jalovaara et al 2022). These differences have been attributed to Finland's skewed national and local sex ratios related to deaths during WWII, mass emigration, low population density and a larger proportion of women compared to men with tertiary education (Rotkirch and Miettinen 2017). The 'two-child family norm' only seems to hold for the higher educated, and childlessness is higher amongst lower educated men and higher educated women (Jalovaara et al. 2022). Nordic gender equality and work-family reconciliation therefore seems to disproportionately aid the highly educated groups to reach the two-child norm. Conversely, the fertility patterns of lower educated men and women in Sweden and Finland are more heterogeneous and polarized between childless and higher parity (Jalovaara et al. 2022)."

Second, to explore this, we have performed additional sensitivity analyses to adjust for possible differences in the highest education levels within sibling pairs. Overall, we observed no significant difference compared to the main analysis (**Figure S16; Supplementary Materials p 55**). Because we mainly considered early-life diseases that onset one year prior to the first childbirth of the sibling with children, most siblings were still relatively young and we don't expect large impacts of sibling-specific confounding factors, which might instead manifest in older ages.

Third, we performed additional population-based analyses considering the individual's education level as a covariate and added the following descriptions to **Results (Manuscript p 14)**:

*"When further adjusting for individuals' highest education level, we observed significant changes in ORs for many diseases, in both women and men (**Supplementary Materials p 56**). In women, after adjusting for the individual's education level, several mental and behavioral disorders due to psychoactive substance abuse (e.g., for acute alcohol intoxication, OR reduced from 2.2 [1.9-2.6] to 1.8 [1.5-2.0], P difference=0.04) had weaker associations with childlessness while malignant neoplasm of brain (OR increased from 5.6 [3.7-8.6] to 14.7 [7.5-28.6], P difference=0.02) and personality disorders such as anxious personality disorder (OR increased from 4.1 [2.8-5.8] to 10.2 [5.9-17.9], P difference= 6.5×10^{-3}) showed stronger associations. In men, however, many disease diagnoses had significantly reduced ORs (16 out of 45 disease diagnoses with increased risk of childlessness) after adjusting for the highest education level while obesity (OR increased from 3.2 [2.8-3.7] to 4.4 [3.7-5.2], P differences= 3.5×10^{-3}) and acute pancreatitis (OR increased from 1.9 [1.6-2.3] to 2.5 [2.1-3.0], P differences=0.04) were only disease diagnoses exhibiting stronger associations."*

Regarding the impact of education on childlessness, we think it's an interesting topic but an in-depth investigation is out of the scope of our study and has been studied in depth using register data by others - see Jalovaara et al (2022). However, we added additional tables along with the following descriptions to the main text:

"Among individual with children, a two-child parity was most prevalent in men (47.1%) and women (48.9%), amongst both Finns (44.0%) and Swedes (50.8%) (Table S2). Both women and men with the

lowest educational attainment were more likely to be childless (e.g., in Finland, 24.2% were childless in women and 37.4% in men) than the general population (e.g., in Finland, 18.8% were childless in women and 27.7% in men) (Table S3-S4). When looking across generations, the education level of parents was correlated with an individual's childless status (Table S3-S4). For example, among index persons whose parents had completed the first stage of tertiary education, men were less likely to be childless (e.g., in Finland, 25.3%) compared to the general population (e.g., in Finland, 27.7%) while women behaved in the opposite way (e.g., in Finland, 21.3%, compared to 18.8% in the general population)."

Additionally, we added more details to **Supplementary Materials p 3-5**:

"To understand the extent to which the social characteristics varied amongst childless individuals and individuals with children in the general population, we considered the highest education level and parental education level in both Finns and Swedes and partnership patterns in Finns only (Table S2-S3), depending on data availability.

Overall, in both countries, women were more educated than men (e.g., in Finland, 49.8% of index women completed the first stage of tertiary education while the proportion dropped to 32.5% for index men). Both women and men with the lowest educational attainment (level 0-2; pre-primary education, primary education or first stage of basic education, or lower secondary or second stage of basic education) were more likely to be childless (e.g., in Finland, 24.2% were childless in women and 37.4% in men) than the general population (e.g., in Finland, 18.8% were childless in women and 27.7% in men). Amongst individuals who completed the first stage of tertiary education (levels 5 and 6), men were less likely to be childless (e.g., in Finland, 20.8%) compared to the general population (e.g., in Finland, 27.7%), however, for women, a slightly higher probability of childlessness was observed (e.g., in Finland, 19.2%, compared to 18.8% in the general population).

Across generations, individuals included in these analyses were more educated than their parents (e.g., in Finland, 42.1% of index persons completed the first stage of tertiary education while the proportion dropped to 20.5% in their parents). The education level of parents was also correlated with an individual's childless status. For example, men who had parents with the lowest educational attainment (level 0-2) were more likely to be childless (e.g., in Finland, 28.9%) than the general population (e.g., in Finland, 27.7%). Among index persons whose parents had completed the first stage of tertiary education (levels 5-6), men were less likely to be childless (e.g., in Finland, 25.3%) compared to the general population (e.g., in Finland, 27.7%) while women behaved in the opposite way (e.g., in Finland, 21.3%, compared to 18.8% in the general population)."

25. The results indicated that mental-behavior disorders and congenital anomalies had the strongest relationship with childlessness. However, congenital anomalies - per the supplement - could only be assessed in the Finnish population. Could the authors shed light on the reasons for this? How should we interpret the result, given the differences in rates of childlessness underlying the two national populations?

Answer: Due to data access limitations, we were only able to analyze congenital anomalies in Finland. However, for congenital anomalies, Sweden has parallel nationwide registration systems as Finland (Arbour L et al., 2009). Given the consistency of disease effects observed for other disease categories, we don't expect that the effects of congenital anomalies on childlessness would differ between the two countries.

26. I am left to wonder about the impact of having an ill sibling. There is substantial literature highlighting the impact one ill child can pose on the family unit as a whole (e.g., in terms of caretaking responsibilities, availability of parental support, familial economic stability, likelihood of divorce, etc. - all features which may factor into siblings' own views on childlessness). There is, further, evidence to suggest that female children are more likely to be given caretaking and other responsibilities in these scenarios, regardless of whether the affected sibling is male or female. It would be interesting to see a familial analysis which included all siblings and/or permitted dissection of associations across gender (e.g., to see whether having an affected male sibling, for instance, had an association with childlessness among unaffected female siblings).

Answer: We think the reviewer raised very good research questions about the potential influence of family members such as the gender of the sick sibling. However, given that not that many disease diagnoses show sex-specific effects from our main analyses (**Figure 3A; Manuscript p 10**), we expect the impact from a sick sibling's gender could be not that large as well.

Also, we think a systematic assessment of childless conditions in unaffected siblings is a research topic requiring different study designs. Therefore, we suggest future studies on associations between diseases and fertility to target more on this topic.

We considered this as one of the limitations of our study and added the following context to the **Discussion** section (**Manuscript p 18-19**) accordingly:

“For example, although full siblings share roughly half of their genetic materials and similar family of origin characteristics and only siblings with the closest birth order were considered, siblings can still experience different childhood conditions and even be influenced by each other’s conditions⁵⁵.”

27. The discussion is severely lacking in references to support key statements, and even full paragraphs? Further, there are interpretive statements in the results that should be necessitate inclusion (and expansion) in the discussion (e.g., the role of sterilization, the reality of reproductive danger in certain disorders).

Answer: We have now extensively rewritten both the introduction and the discussion to more accurately embed it into the existing literature.

The reviewer is also correct that we needed to more explicitly link our findings to the forced sterilization laws in Finland and Sweden. We have now added this paragraph in the discussion in **Manuscript p 18**.

“Our study also reveals the legacy of a troubled and unjust past of reproductive rights, often stemming from eugenic thinking and discrimination related disability and gender^{38,49,50}. We found that intellectual disability was the condition with the strongest association in men, but also high levels of childlessness associated with cerebral palsy and behavioural and personality disorders. Until 1970 in Finland and up to 1976 in Sweden, individuals categorized as having severe mental disorders or physical disabilities – particularly women – were subjected to involuntary sterilization³⁸. We also acknowledge that women with existing diseases, such as women with heart disease, can experience serious complications from pregnancy, thereby noting that strategies for high-risk individuals need further development⁵¹.”

Smaller Items:

28. Line 135 missing 'and'

Answer: We thank the reviewer for the suggestion and have revised the sentence as suggested.

29. Fig 3C. The legend obscures the CIs/elder data points for Obesity

Answer: Thanks for pointing this out! We have changed the legend position for obesity in **Figure 3C (Manuscript p 11)**.

30. Also in 3C: I wonder about the selection of Anorexia here, given there are insufficient figures to conduct the male estimates?

Answer: We thank the reviewer for the suggestion and have removed the figure for Anorexia from **Figure 3C (Manuscript p 11)**.

31. Line 179: I question the assertion that malformations of the digestive and musculoskeletal system can be said to be meaningfully 'more strongly associated with childlessness in women than men,' given the overlapping CIs for the former (and the near overlap on the latter).

Line 188: Again, the CIs overlap for diabetes; and while the interpretation is important, perhaps this belongs in the discussion alongside a wider discussion of the differential causes underlying the signaled associations a various diseases with childlessness.

Answer: Regarding the sex-specific effects observed for malformations of the digestive and musculoskeletal system and also diabetes, we performed standard significance tests to examine whether the point estimates obtained from the two sexes were statistically different from each other and provided the corresponding P values for the effect size difference.

We noticed that, when comparing the two means, there have been extensive discussions regarding the choices of "overlapping confidence intervals" versus "standard statistical tests". Although examining "overlapping confidence intervals" is simple and convenient, previous studies generally suggested that "standard statistical tests" is the more formal option and can avoid extremely conservative comparisons (Schenker and Gentleman, 2001; Payton et al., 2003; Cumming et al., 2009). Therefore, we prefer to continue using our original settings, "standard statistical tests", for sex-specific comparisons.

32. Line 237: what does 'TFR' mean? Is it defined in text?

Answer: We thank the reviewer for the comment and have included both full name and definition of TFR to the **Results (Manuscript p 13)**:

"The completed total fertility rate (TFR), partitioned as the sum of disease and non-disease age-specific fertility rates for each sex, was markedly lower among individuals who carry any of the disease diagnoses significantly associated with childlessness (TFR=1.5 for women and 1.1 for men) compared to the unaffected individuals (2.0 for women and 1.8 for men), resulting in an overall TFR reduction of 38% in women and 21% in men (Table S15-S16)."

Reviewer #3:

33. Key results: Please summarise what you consider to be the outstanding features of the work.

In this study, the authors use comprehensive Finnish and Swedish registry data to examine the relationship between major diseases and childlessness. Using a twin design, the authors show that various major diseases are indeed related to childlessness. For some diseases, their findings echo previous studies, but for other major diseases, this paper is the first to show a relationship with childlessness. The authors have included several valuable additional analyses, e.g. on the age of diagnosis. All in all, I consider this paper a key contribution to the literature on the relationship between health and childlessness.

34. Validity: Does the manuscript have flaws which should prohibit its publication? If so, please provide details.

No, this is a meticulously planned, executed and written up study, and I could not identify any flaws that would limit the paper's contribution to the literature or invalidate its key conclusions.

35. Originality and significance: What are the major claims of the paper? Do you think that they represent a significant advance in the field? If the conclusions are not original, please provide relevant references. On a more subjective note, do you feel that the results presented are of immediate interest to many people in your own discipline, and/or to people from several disciplines?

The main claims of the paper are around the association between various major diseases and childlessness. I see the contributions of this paper mostly in terms of evidence-based advance; conceptually, the link between major diseases (and health more generally) and childlessness is not new, and the twin design that has been used here has also already been applied in earlier work. Nonetheless, I would say that the evidence-based advance is significant, especially because the authors have not just confirmed associations between diseases and childlessness that were already established in earlier work, but have also established relationships between diseases and childlessness that were not identified yet in previous research. Moreover, given the combination of comprehensive registry data and the twin design, the relationships as established here are more robust than most earlier work.

36. Data & methodology: Please comment on the validity of the approach, quality of the data and quality of presentation. Please note that we expect our reviewers to review all data, including any extended data and supplementary information. Is the reporting of data and methodology sufficiently detailed and transparent to enable reproducing the results?

The comprehensive registry data are of excellent quality, the data are presented clearly, and the reporting of the data and methodology is sufficiently detailed and transparent to allow

reproducing the findings.

37. Preregistration: If any part of the work reported in the manuscript was pre-registered, did the authors follow their preregistration plan? Did they report any deviations from their preregistration? Note that we ask authors to provide a link to the pre-registration in the Methods section and state the date of pre-registration. We also ask that authors disclose all deviations from the pre-registered protocol and explain the rationale for deviation (e.g., flaw, suboptimality, or reviewer/editorial request). In cases of deviation from the analysis plan, the originally planned analyses need to be reported in Supplementary Information.

Not applicable.

38. Appropriate use of statistics and treatment of uncertainties: Please include in your report a specific comment on the appropriateness of any statistical tests, and the accuracy of the description of any error bars and probability values.

The statistical tests used are all appropriate given the data and research questions, and issues around error and statistical significance are dealt with sufficiently.

39. Custom code: If the work includes custom code, does the code run as intended? If you are unable to access the code, please contact us.

Not applicable.

40. Conclusions: Do you find that the conclusions and data interpretation are robust, valid and reliable?

Yes, the conclusions are supported by the results, and the presentation and interpretation of the data and the findings support the robustness, validity and reliability of the data.

41. Suggested improvements: Please list additional analyses, experiments or data that could help strengthening the work in a revision.

I have only one suggested improvement of the data. Given the focus on people with siblings, I wondered to what extent the conclusions can be generalized to the population level (after all, not having siblings may also indicate e.g. hereditary issues with fertility). The addition of the population-based analyses in the paper have significantly reassured me that this issue does not affect the main conclusions. However, I would recommend adding a few lines to the results and/or discussion section that reflect on to what extent some specific results (e.g. the estimates of the numbers of people who remained childless because of major diseases) might be slightly biased due to the focus on people with siblings in the main analyses.

Answer: We appreciate the reviewer's recognition of the value of our work.

Regarding the generalizability of sibling-based analysis results to the population level, we think the bias due to the selection of individuals with siblings could be limited in general.

First, there was limited difference in childless proportions amongst people with siblings and people without siblings, as described in the **Results (Manuscript p 7)**:

"The proportion of childless individuals – those with no live-born children – was higher in men (25.4%) than women (16.6%), higher amongst Finns (23.3%) than Swedes (18.7%), and slightly higher in the general population than in individuals who had same-sex full-siblings (0.9% and 0.5% higher in women and men, respectively)."

Second, the estimates from population-based analyses were generally consistent with those obtained from the sibling-based analyses, as described in **Results (Manuscript p 15)**:

"Overall, the population-based design resulted in larger sample sizes, similar ORs, and smaller CIs compared to the sibling-based design (Figure S11)."

However, as suggested by the reviewer, we have added the following contexts to the manuscript to indicate that there could still be slight bias for specific results:

For **Results (Manuscript p 13)**:

"We note that the estimates of the number of children not born or 'missing' due to disease diagnoses could be slightly biased because of the use of only individuals with same-sex full siblings in our main analyses."

And for **Discussion (Manuscript p 19)**:

"Also, the use of only individuals with same-sex full siblings may introduce selection bias, although for most disease diagnoses their effects obtained from sibling-based analyses were not significantly different from those obtained from population-based analyses."

42. References: Does this manuscript reference previous literature appropriately? If not, what references should be included or excluded?

Yes, to me the manuscript does justice to the existing literature on this topic.

43. Clarity and context: Is the abstract clear, accessible? Are abstract, introduction and conclusions appropriate?

Yes, the abstract is clear and easy to follow, and the abstract, introduction and conclusions are appropriate in light of the approach and findings of the study.

44. Please indicate any particular part of the manuscript, data, or analyses that you feel is outside the scope of your expertise, or that you were unable to assess fully.

Not applicable.

Decision Letter, first revision:

22nd March 2023

Dear Dr Ganna,

Thank you once again for your manuscript, entitled "The relationship of early-life diseases with lifetime childlessness: Evidence from Finland and Sweden," and for your patience during the peer review process.

Your manuscript has now been evaluated by 2 reviewers, whose comments are included at the end of this letter. Although the reviewers find your work to be of interest, they also raise some important concerns. We are interested in the possibility of publishing your study in *Nature Human Behaviour*, but would like to consider your response to these concerns in the form of a revised manuscript before we make a decision on publication.

In your revision, we ask that you again respond to all reviewers' concerns. Please follow Reviewer #2 guidance on non-disease related factors, including a multivariate analysis integrating non-disease factors, and the integration of premature mortality and partnership status in your main analyses.

In sum, we invite you to revise your manuscript taking into account all reviewer and editor comments. We are committed to providing a fair and constructive peer-review process. Do not hesitate to contact us if there are specific requests from the reviewers that you believe are technically impossible or unlikely to yield a meaningful outcome.

We hope to receive your revised manuscript within two months. I would be grateful if you could contact us as soon as possible if you foresee difficulties with meeting this target resubmission date.

- Include a "Response to the editors and reviewers" document detailing, point-by-point, how you addressed each editor and referee comment. If no action was taken to address a point, you must provide a compelling argument. When formatting this document, please respond to each reviewer comment individually, including the full text of the reviewer comment verbatim followed by your response to the individual point. This response will be used by the editors to evaluate your revision and sent back to the reviewers along with the revised manuscript.
- Highlight all changes made to your manuscript or provide us with a version that tracks changes.

[REDACTED]

We look forward to seeing the revised manuscript and thank you for the opportunity to review your work. Please do not hesitate to contact me if you have any questions or would like to discuss these revisions further.

Sincerely,

Arunas Radzvilavicius, PhD
Senior Editor, Nature Human Behaviour
Nature Research

REVIEWER COMMENTS:

Reviewer #1:

Remarks to the Author:

Thank you for giving me the chance to review the revised version of the paper. I am very impressed with your attention to detail in addressing the comments made by myself and the other reviewers, and find that most major points raised are well addressed. The definition of childlessness, addition of text box etc are very good.

I just have these final comments/suggestions, which are mainly text edits / additional nuance
- Abstract: p2 line 32 when I read the abstract, I was not sure of the grammatical accuracy of "Around a quarter of childless women do that voluntary..." In this position, voluntary should be an adverb 'voluntarily'

- As one of the other reviewers noted, its sometimes challenging to measure voluntary childlessness accurately, as people often rationalise their situational constraints as choices. Maybe some acknowledgment of the difficulties in measuring this, in the text box, would be good (e.g. although there are measurement issues...)

Intro; I am not sure I understand what the phrase 'permanent childlessness' means (especially when you consider what 'temporary' childlessness might be)- I think the more demographic way to put it would be 'lifetime childlessness'

- p4 line 108-109 the fact that a large proportion of individuals are still involuntarily childless suggests that biology and diseases likely have a considerable influence

I think this might be too strong an assumption... what about other related factors like difficulties finding the right partner, etc (this would also count as involuntary childlessness)

- p 5 line 117 . I think you missed something in this sentence: Finally, many diseases are associated with lower socioeconomic status, unemployment, and economic uncertainty, which may amplify the psychosocial impact of diseases on childlessness- what about how it affects propensity to be in a stable partnership? This is a crucial prerequisite to fertility in most cases.

- The stratification by parity results raise the question for me, of whether those same diseases associated with childlessness also help us to understand / predict low parity (i.e. just one child), which is quite rare and selected state in the Nordic countries. There is a small body of work on only children and their worse health outcomes in these countries. Some of this is explained by lower SES but there is a remaining portion of poor health unexplained. Maybe parent's poorer health status and these conditions you identify could help explain those results. Something perhaps to reflect on the discussion...

p 17 | 435- there are no references or citations to back up these additional statements you have added.

Otherwise, I'd like to congratulate the authors on an excellent revision

Reviewer #2:

Narratively, this revision is much improved and I thank the authors for their evident effort to include more nuanced coverage (particularly in the introduction) of childlessness, and the social factors that may shape this outcome. However, in doing so, the authors signal what remains, for me, a challenge in understanding the intention/significance of the work: the reality that, given the intense bias against childless persons (particularly women), and the cultural context surrounding persons in the study populations (again, particularly women; the authors reference the 'psychological, physical and sexual violence, and economic coercion' around the topic), it seems to me inescapable that the proportion who would identify as 'voluntarily childless' in the narrow way it has been framed here (e.g., always knew they did not want children) would be almost nil (5% as estimated from the paper's literature). The definition of 'involuntary,' by contrast, is quite expansive - including, by default, any persons whose childlessness is the result of the constellation of factors that may lead to the 'repeated postponement' referenced. Among these, of course, being the topic of the study: one's disease status.

The challenge is: if, as stated, the goal of the work is to understand the relationship of this singular factor to 'voluntary childlessness,' there needs to be a sincere effort to distinguish the role of disease from the myriad other factors with demonstrable impacts on both partnering success and child bearing. Some of these are basic and immutable: individuals need to survive long enough to partner and reproduce, and when they attain this age need to be able to find a partner (in most cases, at least). Others have measurable import, but exploration would be needed to understand how they interact with disease in the (proposed) causal relationship between disease and childlessness: education status is a good example, but certainly not the only one (e.g., employment status, socioeconomic status, parental health, region of residence, spirituality/religion, etc.). However, in the manuscript at hand, these features are never approached in the multivariate manner needed to meaningfully dissect this causal complexity. Premature mortality (e.g., death before follow-up) is, for instance, not integrated in the main results. Partnership status, likewise, is not explored until late in the manuscript (where it is shown to account for a substantial proportion of the overall effect, particularly among men, and neutralizes the relationship of a meaningful proportion of individual disease associations). And review of the supplementary material suggests no attempt to conduct a multivariate analyses, integrating these features with other important covariates (e.g., education status) in a way that would allow parsing of the role of disease relative these clearly impactful features.

While I respect the scale of the descriptive elements of this work, and acknowledge the efforts of the authors to add depth to their coverage, without these analytic efforts, I find there is a lack of necessary nuance in these findings. To the degree that I have concern that the work will feed the bias against childless individuals, rather than aid in understanding the features (including disease) which may drive this outcome - be it truly 'involuntary' or not.

I see two possible pathways here, as the authors think about what to do next with this paper: one option would be to reshape the manuscript, as it is, into a more simple and descriptive 'encyclopedia of risk,' wherein just the basic descriptive numbers are provided and then the discussion expanded to address the pathways through which these relationships may be emerging. The alternative, in my view, would be to stay true to the stated focus of this paper, and therefore augment the analyses to allow multivariate examination and sincere parsing of the relative role of disease to other factors - shown in the univariate examinations already done - to have clear impact on risk for childlessness.

Two standalone comments:

1. The disease-specific age of onset analyses appears very underpowered (>5 is a very low bar, in terms of N, for computing these effects and the intervals are too large for the trend to be interpretable). Suggest removal and focus on the overall effect. I acknowledge such an approach may be necessary for the suggested multivariate analyses as well.
2. It would be helpful to see some figures re: 'registered partnership,' in these Scandinavian

samples. My understanding is that such registration is relatively rare (marriage is not a given for couples, for instance). I do wonder about the differences in partnership being influenced by this.

Author Rebuttal, first revision:

Reviewer #1:

Thank you for giving me the chance to review the revised version of the paper. I am very impressed with your attention to detail in addressing the comments made by myself and the other reviewers, and find that most major points raised are well addressed. The definition of childlessness, addition of text box etc are very good. I just have these final comments/suggestions, which are mainly text edits/additional nuance.

1. Abstract: p2 line 32 when I read the abstract, I was not sure of the grammatical accuracy of "Around a quarter of childless women do that voluntary..." In this position, voluntary should be an adverb 'voluntarily'

Answer: We thank the reviewer for recognizing our major revisions and are gratified that the reviewer considers that most points are well addressed. We appreciate your pointing out this inadvertent error. We have now gone through the entire paper and fixed sentences with similar grammar errors.

First, on **Manuscript p2:**

"Around a quarter of childless women do that voluntarily, suggesting a remaining role for disease."

Second, on **Manuscript p3:**

"it has been estimated that around a quarter of the current 15-20% women who are childless in Europe are voluntarily childless³."

Third, on **Manuscript p5:**

"A systematic review found that up to one-third of IBD patients opted to be voluntarily childless, often related to lack of knowledge related to pregnancy-related IBD issues²⁹, with clinical studies finding that women with moderate to severe IBD had higher incidences of adverse pregnancy outcomes³⁰."

Fourth, on **Manuscript p18:**

"Given that estimates of the voluntarily childfree group are around a quarter of total childlessness, we note that public health interventions should take this group of voluntarily childfree individuals into account."

2. As one of the other reviewers noted, it's sometimes challenging to measure voluntary childlessness accurately, as people often rationalise their situational constraints as choices. Maybe some acknowledgment of the difficulties in measuring this, in the text box, would be good (e.g. although there are measurement issues...)

Answer: We thank the reviewer for the excellent suggestion. We have now acknowledged the

difficulties in measurement issues when referencing the proportion of voluntary versus involuntary childlessness, which indeed has been a major area of research in the literature (**Manuscript p3, Text Box**):

“Although researchers have engaged in in-depth studies showing the measurement challenges of differentiating the two, it has been estimated that around a quarter of the current 15-20% of women who are childless in Europe are voluntarily childless³.”

3. Intro; I am not sure I understand what the phrase 'permanent childlessness' means (especially when you consider what 'temporary' childlessness might be)- I think the more demographic way to put it would be 'lifetime childlessness'

Answer: This is an excellent point and we thank the reviewer and have now changed “permanent childlessness” to “lifetime childlessness” (**Manuscript p3**):

“In Western European countries, around 20% of women born around 1965 remained childless, with the highest levels of lifetime childlessness now in East Asia, ranging from 28% (Japan) to 35% (Hong Kong) for women born in 1975^{7,8}.”

4. p4 line 108-109 the fact that a large proportion of individuals are still involuntarily childless suggests that biology and diseases likely have a considerable influence. I think this might be too strong an assumption... what about other related factors like difficulties finding the right partner, etc (this would also count as involuntary childlessness).

Answer: We thank the reviewer for the comment and have rephrased the sentence as suggested (**Manuscript p4**):

“While the aforementioned demographic and socioeconomic literature has shown social, economic, and structural factors as having a significant effect on childlessness, the fact that a large proportion of individuals experience this involuntarily suggests that diseases might play a role in influencing individuals' chances of being childless or having a particular number of children over their lifetime (Figure S1).”

5. p 5 line 117. I think you missed something in this sentence: Finally, many diseases are associated with lower socioeconomic status, unemployment, and economic uncertainty, which may amplify the psychosocial impact of diseases on childlessness- what about how it affects propensity to be in a stable partnership? This is a crucial prerequisite to fertility in most cases.

Answer: We agree with the reviewer that partnership stability can be one of the major factors mediating the psychosocial impact of diseases on childlessness. To clarify this, we have rephrased the sentence as follows (**Manuscript p5**):

“Finally, many diseases are associated with lower socioeconomic status, unemployment, and economic uncertainty, which may amplify the psychosocial impact of diseases on childlessness, possibly by increasing partnership instability or reducing fertility intentions²⁵.”

6. The stratification by parity results raise the question for me, of whether those same diseases associated with childlessness also help us to understand / predict low parity (i.e. just one child), which is quite rare and selected state in the Nordic countries. There is a small body of work on only children and their worse health outcomes in these countries. Some of this is explained by lower SES but there is a remaining portion of poor health unexplained. Maybe parent's poorer health status and these conditions you identify could help explain those results. Something perhaps to reflect on the discussion...

Answer: We thank the reviewer for the excellent suggestion and have now added the following discussion to the manuscript (**Manuscript p16**):

"Additionally, in terms of risks for childlessness-associated early-life diseases, childless individuals are more similar to their siblings with one child rather than those with higher parities. This suggests that the disease-childlessness associations identified from our main analyses can also be used to understand the overall impact of parental health status on low parity (i.e., one child).

7. p 17 | 435- there are no references or citations to back up these additional statements you have added.

Answer: Thank you for these comments. We now make it more explicit that the statements are findings from the current study and that we extrapolate from previous studies that the effects of disease diagnoses on partnership are likely mediated by socio-cultural factors (**Manuscript p16**):

"The effects of disease diagnoses on partnership formation are likely mediated by complex socio-cultural factors, social norms, and sex-specific behavioural preferences^{45,46,47}. For example, in the current study, we find that mental-behavioural disorders such as schizophrenia and acute alcohol intoxication are strongly associated with singlehood in men. The effects are substantially smaller in women, which likely reflects how different social and individual preferences for behavioural phenotypes correlated to mental-behavioural disorders impact partner choice in the two sexes⁴⁶."

8. Otherwise, I'd like to congratulate the authors on an excellent revision.

Answer: We really appreciate the very insightful comments given by the reviewer. We also want to thank the reviewer for the appreciation of the quality of our work and for recognizing the efforts we put into the revision.

Reviewer #2:

9. Narratively, this revision is much improved and I thank the authors for their evident effort to include more nuanced coverage (particularly in the introduction) of childlessness, and the social factors that may shape this outcome. However, in doing so, the authors signal what remains, for me, a challenge in understanding the intention/significance of the work: the reality that, given the intense bias against childless persons (particularly women), and the cultural context surrounding persons in the study populations (again, particularly women; the authors reference the 'psychological, physical and sexual violence, and economic coercion' around the topic), it seems to me inescapable that the proportion who would identify as

'voluntarily childless' in the narrow way it has been framed here (e.g., always knew they did not want children) would be almost nil (5% as estimated from the paper's literature). The definition of 'involuntary,' by contrast, is quite expansive - including, by default, any persons whose childlessness is the result of the constellation of factors that may lead to the 'repeated postponement' referenced. Among these, of course, being the topic of the study: one's disease status.

Answer: We thank the reviewer for recognizing our major revisions of the paper and repositioning it more broadly to encompass the rich social and cultural literature within this area of research. We note that the reviewer lists mainly two points above which we will address one by one.

Firstly, the reviewer reiterates initial concerns about stigmatisation of childless persons and particularly women. We would like to draw the reviewer's attention to our very explicit addressing of this issue in the **Text box**. We also note that the majority of literature on infertile women experiencing violence and economic coercion is from middle and low-income countries (**Manuscript p 3**):

"Childless individuals are subjected to discrimination and marginalisation in many societies⁴, particularly in middle and low-income countries, where infertile women experience multiple types of violence and coercion⁵. A parallel line of work, which is not the position of this paper or authors, is to problematize and stigmatize childless individuals as egoistic and place blame on this group for producing a so-called 'demographic disaster' of shrinking and ageing populations and collapse of social security systems⁶. The approach of this paper is to provide a neutral, data-driven, and factual examination of early-life diseases related to childlessness, with the aim to design a better understanding of health to prevent childlessness among those who want to have children."

Secondly, regarding the proportion of voluntary childlessness, we want to clarify that the statement "it seems to me inescapable that the proportion who would identify as 'voluntarily childless' in the narrow way it has been framed here (e.g., always knew they did not want children) would be almost nil (5% as estimated from the paper's literature)" is not correct.

The proportion of voluntary childlessness among all individuals of the reproductive age is 4-5% (Miettinen and Szalma, 2014). Given that the overall lifetime childlessness proportion is 15-20% (Sobotka, 2017), we stated that around a quarter of childless individuals are voluntarily childless. We have now rephrased the sentence to avoid any misunderstandings (**Manuscript p2**):

"Around a quarter of childless women do that voluntarily, suggesting a remaining role for disease."

Also, we recognize the reviewer's point about the difficulties in measuring the proportion of voluntary versus involuntary childlessness, we have now explicitly noted measurement challenges in the text (**Manuscript p3**):

"Although researchers have engaged in in-depth studies showing the measurement challenges in differentiating the two, it has been estimated that around a quarter of the current 15-20% of women who are childless in Europe are voluntarily childless³."

10. The challenge is: if, as stated, the goal of the work is to understand the relationship of this singular factor to 'voluntary childlessness,' there needs to be a sincere effort to distinguish the role of disease from the myriad other factors with demonstrable impacts on both partnering success and child bearing. Some of these are basic and immutable: individuals need to survive long enough to partner and reproduce, and when they attain this age need to be able to find a partner (in most cases, at least).

Answer: Before providing a detailed answer, we think there is a spelling mistake in this statement: "The challenge is: if, as stated, the goal of the work is to understand the relationship of this singular factor to 'voluntary childlessness,'" and we instead interpret it as "as stated, the goal of the work is to understand the relationship of this singular factor to 'involuntary childlessness'". We hope to be correct in doing so.

We note that the reviewer suggests two issues here, premature mortality and partnership status, which we will address one by one.

Firstly, the reviewer notes that premature mortality needs to be integrated into the main results. Given that we are looking at Finland and Sweden, which have very low mortality rates for individuals younger than age 16, this integration is not meaningful for our analyses. However, to clarify this point to the reviewer and editor we present here the age-specific mortality rates for these countries to assuage any concerns.

Since we don't have access to mortality data of individuals under age 16 in the studying Finnish and Swedish populations, we cite the mortality rates documented in publicly available data resources.

Based on the available data on infant mortality rate (IMR) (**Table R1**) and under-five mortality rate (U5MR) (**Figure R1**) in Finland and Sweden, it is evident that both countries experienced significantly lower IMR and U5MR compared to the world average between 1956 and 1968. While historical data on mortality rates for children under 16 years of age is not readily available, it is generally understood that children under five are the most vulnerable, and the mortality rate between 5-16 is lower than U5MR. Therefore, we can reasonably assume that premature mortality rates of individuals prior to reproductive age in Finland and Sweden during this period were low and not a major concern for our study.

Table R1. Comparative Infant Mortality Rates (IMR) in Finland, Sweden, and the World (1956-1968).

Year	IMR Finland	IMR Sweden	IMR World
1956	28.31	17.85	132.46
1957	26.72	17.34	130.09
1958	25.13	16.84	127.72
1959	23.93	16.54	126.25
1960	22.74	16.25	124.77
1961	21.55	15.95	123.30
1962	20.35	15.66	121.83
1963	19.16	15.36	120.35
1964	18.25	14.76	117.11
1965	17.34	14.15	113.86
1966	16.43	13.54	110.62
1967	15.524	12.94	107.38
1968	14.616	12.33	104.13

Note: Infant mortality rate (IMR) is a statistical measure that represents the number of deaths of infants under one year of age per 1,000 live births in a given population during a specific period, usually a year. Data source:

United Nations, Department of Economic and Social Affairs, Population Division (2022).
World Population Prospects 2022: Data Sources. (UN DESA/POP/2022/DC/NO. 9).

Figure R1. Child mortality in Finland, Sweden, and other countries, 1956 to 1968.

Note: the child mortality rate shows the share of children (born alive) who die before they are five years old.
Data source: Gapminder (2017) & UN IGME (2018) from <https://ourworldindata.org/child-mortality>

Additionally, in the method section, we included the following statement when describing the study population (**Manuscript p24**):

“We further excluded individuals who died before the age of 16 to eliminate the impact of diseases on pre-reproductive survival.”

Secondly, we agree with the reviewer that the inability to find a partner is an important mediator of childlessness. This is also the main reason why we have an entire section of “mediation effect by singlehood” and consider this as one of the strengths of our paper.

Specifically, we performed two sets of analyses:

- (a) a mediation analysis to quantify the mediating effect of singlehood on the observed disease-childlessness associations;
- (b) an additional association analysis restricting to individuals with registered partners, to investigate if some of the observed disease-childlessness associations remained significant among partnered individuals.

The detailed descriptions of the abovementioned analyses and the corresponding results can be found at **Manuscript p13** and **p27**, with “Mediation effect by singlehood” as a subtitle.

Also, we included the following discussion at **Manuscript p17**:

“Third, by leveraging the information about family formation and singlehood, we could explore to what extent this information, which has been described as an important mediator of childlessness⁴⁸, mediated the association between disease and childlessness.”

11. Others have measurable import, but exploration would be needed to understand how they interact with disease in the (proposed) causal relationship between disease and childlessness: education status is a good example, but certainly not the only one (e.g., employment status, socioeconomic status, parental health, region of residence, spirituality/religion, etc.). However, in the manuscript at hand, these features are never approached in the multivariate manner needed to meaningfully dissect this causal complexity. Premature mortality (e.g., death before follow-up) is, for instance, not integrated in the main results. Partnership status, likewise, is not explored until late in the manuscript (where it is shown to account for a substantial proportion of the overall effect, particularly among men, and neutralizes the relationship of a meaningful proportion of individual disease associations). And review of the supplementary material suggests no attempt to conduct a multivariate analyses, integrating these features with other important covariates (e.g., education status) in a way that would allow parsing of the role of disease relative these clearly impactful features.

While I respect the scale of the descriptive elements of this work, and acknowledge the efforts of the authors to add depth to their coverage, without these analytic efforts, I find there is a lack of necessary nuance in these findings. To the degree that I have concern that the work will feed the bias against childless individuals, rather than aid in understanding the features (including disease) which may drive this outcome - be it truly 'involuntary' or not.

I see two possible pathways here, as the authors think about what to do next with this paper: one option would be to reshape the manuscript, as it is, into a more simple and descriptive 'encyclopedia of risk,' wherein just the basic descriptive numbers are provided and then the discussion expanded to address the pathways through which these relationships may be emerging. The alternative, in my view, would be to stay true to the stated focus of this paper, and therefore augment the analyses to allow multivariate examination and sincere parsing of the relative role of disease to other factors - shown in the univariate examinations already done - to have clear impact on risk for childlessness.

Answer: We do agree there are many health and socio-economic factors impacting childlessness, which we have referenced and discussed in more detail in the revised version of this study. In this study, we are specifically focusing on early-life diseases. These diseases can causally impact childlessness (either by biologically impacting fertility or by reducing the chance to find a partner) or the association between diseases and childlessness can be confounded. For example, the lower parental socioeconomic status might both increase disease risk and decrease the chance of finding a partner and therefore increase childlessness. One option to account for confounding would be to include measurable confounders in a multivariable model. However, this approach has several challenges including residual confounding and a lack of extensive socio-economic or environmental characteristics. An alternative approach, which many have pointed out as being more powerful, is to use a sibling design, which we have in fact done in this study. Familial confounders (which represent the largest source of confounding in younger individuals) can be better accounted for with this design. As most of the socio-economic information that is available to us is shared between siblings (e.g., the "parental health" conditions proposed by the reviewer are largely shared by siblings, especially for those with close birth orders), there is no need to fit a multivariable model when using a sibling design. We showed, for example, when adjusting for individual's educational attainment (which might vary between siblings), it does not change the results (**Manuscript p15**). In conclusion, we followed the current position in the field, which is that using a sibling design helps account for familial confounding although we acknowledge that remaining confounding might remain (**Manuscript p18**).

One might attempt to fit a multivariable model to explore the effect of mediators (as we did for partnership formation), but we want to highlight that this is not the main scope of the paper and we acknowledge that many mediators on the pathways between disease and childlessness might exist. Such mediators might be explored in studies with a rich collection of behavioural and environmental information to characterize how diseases can impact childlessness.

The scope of this work is to provide a comprehensive view of which diseases can impact childlessness and their relative importance. We think that even without understanding the exact mechanism for which a disease impacts childlessness (but we also doubt this can simply be characterized via multivariable models), our results provide an important compendium for researchers and policymakers and generate new hypotheses that can be explored in detailed follow-up analyses.

Two standalone comments:

12. The disease-specific age of onset analyses appears very underpowered (>5 is a very low bar, in terms of N, for computing these effects and the intervals are too large for the trend to be interpretable). Suggest removal and focus on the overall effect. I acknowledge such an approach may be necessary for the suggested multivariate analyses as well.

Answer: We agree with the reviewer that the power of statistical test is depending on the sample size. However, >5 disease cases (within a certain age group) is not a “very low bar” especially for a sibling design which already limited the analyses to sibling pairs discordant on both disease status and childlessness outcome. Furthermore, the onset age is considered as fixed effect in a mixed model, thus, the “>5 disease case” requirement is for minimum N of observations for a specific “level”. Also, in the result table (**Table S9**), we reported 95% confidence interval and N of disease cases of each included age group to help understand the identified associations.

13. It would be helpful to see some figures re: 'registered partnership,' in these Scandinavian samples. My understanding is that such registration is relatively rare (marriage is not a given for couples, for instance). I do wonder about the differences in partnership being influenced by this.

Answer: Firstly, in **Methods**, we included the following sentences to describe the marriage and partnership data we collected from Finnish and Swedish nationwide population registers (**Manuscript p25**):

“For index individuals in Finland, we obtained their longitudinal marriage and partnership information between 11 April 1971 and 31 December 2018 from the Population Information System. In Sweden, we collected information of married couples and cohabiting unions with biological children between 1 January 1977 and 31 December 2017 from Statistics Sweden. We defined individuals as partnerless if they did not have any abovementioned marriage or partnership registered by age 45 (women) or 50 (men).”

Secondly, depending on data availability, we also included descriptive statistics of partnership

patterns for the studying Finnish population in our previous revision (**Supplementary Materials p8, Table S3**).

Table S3: Descriptive statistics of social characteristics for index persons in Finland.

Variable	Women		Men	
	With children	Childless	With children	Childless
Partnership pattern				
No partner	69,263 (14.9%)	70,300 (65.1%)	45,308 (13.5%)	93,067 (72.6%)
Ever-married	395,127 (85.0%)	37,169 (34.4)	289,834 (86.5%)	34,719 (27.1%)
Registered partnership	215 (0.05%)	444 (0.41%)	55 (0.02%)	427 (0.33%)

Note: this is not the full version of **Table S3**, only the descriptive statistics related to partnership pattern were copied here.

Decision Letter, second revision:

20th April 2023

Dear Dr Ganna,

Thank you for submitting your revised manuscript, "The relationship of early-life diseases with lifetime childlessness: Evidence from Finland and Sweden". After careful consideration and discussion with my colleagues, I am sorry to have to tell you that we do not feel that the referees' comments have been sufficiently addressed to justify sending this revision back for peer review or acceptance.

In order to consider this manuscript further we would request that you please do your best to fully address all of the comments of the reviewers. In particular, in the previous letter we asked that you include "a multivariate analysis integrating non-disease factors, and the integration of premature mortality and partnership status in your main analyses" as per reviewer's request.

Should you be able to adequately respond to these and the reviewers' other concerns, we would be happy to look at a revised manuscript. We shall hope to receive your revised version within a month. Do let me know if you are unable to add the requested analyses so we could withdraw your submission.

Please use the link below to submit a suitably revised manuscript and updated response to referees when they are ready.

[REDACTED]

Sincerely,

Arunas Radzvilavicius, PhD
Senior Editor, Nature Human Behaviour
Nature Research

Decision Letter, third revision:

8th September 2023

Dear Dr. Ganna,

Thank you for your patience as we've prepared the guidelines for final submission of your Nature Human Behaviour manuscript, "The relationship of early-life diseases with lifetime childlessness: Evidence from Finland and Sweden" (NATHUMBEHAV-22082041C). Please carefully follow the step-by-step instructions provided in the attached file, and add a response in each row of the table to indicate the changes that you have made. Please also check and comment on any additional marked-up edits we have proposed within the text. Ensuring that each point is addressed will help to ensure that your revised manuscript can be swiftly handed over to our production team.

We would hope to receive your revised paper, with all of the requested files and forms within two-three weeks. Please get in contact with us if you anticipate delays.

Nature Human Behaviour offers a Transparent Peer Review option for new original research manuscripts submitted after December 1st, 2019. As part of this initiative, we encourage our authors to support increased transparency into the peer review process by agreeing to have the reviewer comments, author rebuttal letters, and editorial decision letters published as a Supplementary item. When you submit your final files please clearly state in your cover letter whether or not you would like to participate in this initiative. Please note that failure to state your preference will result in delays in accepting your manuscript for publication.

In recognition of the time and expertise our reviewers provide to Nature Human Behaviour's editorial process, we would like to formally acknowledge their contribution to the external peer review of your manuscript entitled "The relationship of early-life diseases with lifetime childlessness: Evidence from Finland and Sweden". For those reviewers who give their assent, we will be publishing their names alongside the published article.

Cover suggestions

We welcome submissions of artwork for consideration for our cover. For more information, please see our https://www.nature.com/documents/Nature_covers_author_guide.pdf target="new"> guide for cover artwork.

ORCID

Non-corresponding authors do not have to link their ORCIDs but are encouraged to do so. Please note that it will not be possible to add/modify ORCIDs at proof. Thus, please let your co-authors know that if they wish to have their ORCID added to the paper they must follow the procedure described in the following link prior to acceptance:

Nature Human Behaviour has now transitioned to a unified Rights Collection system which will allow our Author Services team to quickly and easily collect the rights and permissions required to publish your work. Approximately 10 days after your paper is formally accepted, you will receive an email in providing you with a link to complete the grant of rights. If your paper is eligible for Open Access, our Author Services team will also be in touch regarding any additional information that may be required to arrange payment for your article.

Please note that *Nature Human Behaviour* is a Transformative Journal (TJ). Authors may publish their research with us through the traditional subscription access route or make their paper immediately open access through payment of an article-processing charge (APC). Authors will not be required to make a final decision about access to their article until it has been accepted. Find out more about Transformative Journals

[REDACTED]

Best regards,
Alex McKay
Editorial Assistant
Nature Human Behaviour

On behalf of

Arunas Radzvilavicius, PhD
Senior Editor, Nature Human Behaviour
Nature Research

Reviewer #2:

Remarks to the Author:

I accept the revisions and thank the authors for their attention to my comments on this sensitive piece.

Author Rebuttal, third revision:

Point-to-point response to the editors' comments

1. Title, the editor suggested to change the title from "The relationship of early-life diseases with lifetime childlessness: Evidence from Finland and Sweden" to "Evidence from Finland and Sweden on the relationship between early-life diseases and lifetime involuntary childlessness".

Answer: We thank the editor for the suggestion of a new title.

Firstly, regarding the change of "childlessness" to "involuntary childlessness", we agree with the editor that, ideally, the association of interests would be diseases and involuntary childlessness. However, due to the measurement challenges of differentiating voluntary and involuntary childlessness which we outline in considerable detail, it is more accurate to state that our study considered both conditions. To avoid possible misinterpretation of the results, it would be more in line with the literature to keep the more general term "childlessness".

Secondly, to make clear that our focus is on both men and women, we have added this information to the title.

The new title is:

"Evidence from Finland and Sweden on the relationship between early-life diseases and lifetime childlessness in men and women"

2. As it is currently written, your abstract, introduction and text box are heavily focused on women, whereas your work examines childlessness and disease for both men and women. This framing may inadvertently communicate that involuntary childlessness is women's "problem" only (and conversely, voluntary childlessness the prerogative of women only). We would suggest a more balanced framing that discusses childlessness for both sexes. (The fact that comparatively little research has been carried out in men can be mentioned up-front.) After revisions, please ensure that your abstract does not exceed 150 words.

Answer: We thank the editor for this observation and have now highlighted the study of men in the title, introduction, and throughout the paper. The literature review was indeed more detailed on the conditions in women due to the fact that many societal shifts such as educational expansion and entry into the labour market linked to childlessness was largely related to changes in women's lives. But we also acknowledge that in fertility research, there has also been a lack of studies on

men. To balance and rectify this, we have now added some additional sentences that highlight research on men.

For the **Abstract**, to be clear and concise, we have now removed the sentence introducing the percentage of voluntarily childfree women while added a mention of the proportion of childlessness in men. The abstract now has exactly 150 words and reads as (**Manuscript p2, Line 32-33**):

“The percentage of people without children over their lifetime is approximately 25% in men and 20% in women. Individual diseases have been linked to childlessness, mostly in women, yet we lack a comprehensive picture of the effect of early-life diseases on lifetime childlessness. We examined all individuals born 1956-1968 (men) and 1956-1973 (women) in Finland (n=1,035,928) and Sweden (n=1,509,092) to completion of their reproductive lifespan in 2018. Leveraging nationwide registers, we associated sociodemographic and reproductive information with 414 diseases across 16 categories, using a population and matched pair case-control design of siblings discordant for childlessness (71,524 full-sisters, 77,622 full- brothers). The strongest associations were mental-behavioral disorders, particularly amongst men, congenital anomalies and endocrine-nutritional-metabolic disorders, strongest amongst women. We identified novel associations for inflammatory and autoimmune diseases. Associations were dependent on age at onset and mediated by singlehood and education. Evidence can be used to understand how disease contributes to involuntary childlessness.”

For the **Text box**, we have now added references to support the commonplace age thresholds we used when defining lifetime childlessness in both men and women (**Manuscript p2, Line 49**):

“We use the term childlessness to describe individuals that have had no live-born children by the end of their reproductive lifespan (age 45 for women¹⁻³; 50 for men^{2,3}).”

In the **introduction**, we have made the following revisions to present a more balanced framing of both sexes.

We note on **Manuscript p3, Line 76-79** that both men and women face work-life reconciliation challenges, etc. instead of saying couples:

“At the same time, both men and women faced work-life reconciliation constraints when planning to have children including lack of access to childcare, challenging housing conditions, economic uncertainty, inability to find a partner, and the general absence of supportive family policies¹².”

We also note on **Manuscript p3, Lines 83-85** that economic precarity and labour market uncertainty and childlessness are found mostly for men.

“Another shift, which has been particularly linked to men, is the growth of precarious, temporary, and uncertain employment, leading many to postpone or forgo entering partnerships and parenthood^{16,17}.”

To balance the **introduction** to also focus on men, we added the new text to **Manuscript p3, Line 89-94**:

“The rise in childlessness amongst men includes many of the factors above, such as the role of economic uncertainty, socioeconomic circumstances, and structural challenges such as childcare. More recently a key predictor for male childlessness is the rise of multiple and short partnerships¹⁰,

and a normative shift to a growing group of men who are disinterested in becoming a father¹⁹. Others have linked rises in male childlessness to lifestyle factors, such as becoming more sedentary, alcoholism, tobacco use, and poor diet²⁰.”

Regarding the consequences of being childless, we have now mentioned that some conditions are shared among men and women (**Manuscript p4, Line 106-108**):

“From the standpoint of public health, involuntary childlessness may impact other health domains, with childless men and women more likely to suffer from relationship dissolution, lower levels of self-esteem and isolation, and higher risks of clinical depression^{12,25}.”

And, for men, we added the example in men when describing the social discrimination and marginalization people without children may suffer (**Manuscript p4, Line 111-113**):

“For childless men, their workplace interactions and career opportunities have been found to be negatively affected, with psychological wellness rarely addressed²⁵.”

Finally, as kindly suggested by the editor, we have now moved the following sentence up- front (now in the paragraph reviewing the literature linking diseases to childlessness) (**Manuscript p5, Line 139-142**):

“Overall, for men, comparatively limited information – often from highly selective semen samples from infertility clinics – has been collected on the causes and consequences of infertility and having children^{35,36}, despite infertility being a major public health concern.”

3. We recommend that in the context of voluntary choice/preference not to have children, you use the term ‘childfree’ (without quotes), as this is the term that seems to be preferred by this group of people. If you would like to discuss this, please let us know.

Answer: We thank the editor for the thoughtful comment and have now changed the term “voluntary childless” to “child free” (without quotes) through the manuscript.

4. Line 67, the end of their reproductive lifespan (age 45 for women; 50 for men). Please provide references to support these ages as marking the end of reproductive lifespans in women and men. (There seems to be quite a lot of variability in definitions of reproductive lifespan.)

Answer: We thank the editor for pointing this out and have now added relevant references to support the age thresholds we used when defining lifetime childlessness in women and men (Craig et al. 2014; Matthieson et al. 2023; Mills et al. 2021) (**manuscript p2**).

References:

Craig, B.M., Donovan, K.A., Fraenkel, L., Watson, V., Hawley, S. and Quinn, G.P., 2014. A generation of childless women: lessons from the United States. *Women's Health Issues*, 24(1), pp.e21-e27.

Mills, M. C. et al. Identification of 371 genetic variants for age at first sex and birth linked to

externalising behaviour. *Nat. Hum. Behav.* 5, 1717–1730 (2021).

Mathieson, I., Day, F.R., Barban, N., Tropf, F.C., Brazel, D.M., eQTLGen Consortium, BIOS Consortium van Heemst Diana 47, Vaez, A., van Zuydam, N., Bitarello, B.D. and Gardner, E.J., 2023. Genome-wide analysis identifies genetic effects on reproductive success and ongoing natural selection at the FADS locus. *Nature human behaviour*, 7(5), pp.790-801.

5. Line 69, Inability to find a partner is not a biological/fecundity factor. It seems that there are different types of involuntary childlessness, including circumstantial (e.g., inability to find a partner) and social (e.g., poverty). The main text provides relevant information. Please revise this to ensure that reasons/determinants are accurately described.

Answer: We thank the editor for pointing this out and have now improved the sentence as follows (**Manuscript p2**):

“Childlessness is defined in the literature as being voluntary or childfree⁴ (e.g., active choice, preference⁵), involuntary (e.g., infertility, stillbirth, reproductive age mortality), or a mix of the two, such as circumstantial situations related to partnership and socio-economic status (Supplementary Fig. 1).”

The rephrased sentence now also matched what have been presented in the introduction and Supplementary **Fig. 1**.

6. Text box, the editor commented on the following sentence that there is no reason to give so much space/air to non-scientific, low value work. Instead, you can add it as a reference to the preceding sentence: “A parallel line of work, which is not the position of this paper or authors, is to problematize and stigmatize childless individuals as egoistic and place blame on this group for producing a so-called ‘demographic disaster’ of shrinking and ageing populations and collapse of social security systems⁷.”

Answer: We thank the editor for the comment and have now removed the sentence from the manuscript.

7. Line 84, Remained when? Please provide the date when this prevalence was estimated.

Answer: We thank the editor for pointing this out and have now added the year when the prevalence was estimated to the **manuscript (p3)**:

*“In Western European countries, around 20% of women born around 1965 remained childless **by 2010**, with the highest levels of lifetime childlessness now in East Asia, ranging from 28% (Japan) to 35% (Hong Kong) for women born in 1975^{7,8}.”*

8. "A recent systematic review found that infertile women, particularly those in middle and low-income countries, are also more likely to experience psychological, physical, and sexual violence as well as economic coercion⁵". Is this reference correct? Ref 5 is an introduction to a special issue, rather than a systematic review.

Answer: We thank the editor for noticing this. The correct reference we previously want to refer to should be Wang et al., 2022 (PMID: 35561719) and we have now fixed this in the revised manuscript (**p4, Line 111**).

9. If you could edit other references to SI material to match this format, that would be great! Also, in the SI files themselves.

Answer: We have now formatted all references to SI materials, as kindly suggested by the editor.

10. Panel labels for **Fig. 1**, could you make this bold in the figure itself too?

Answer: As suggested by the editor, we have now changed all panel labels of **Fig. 1** to be lower case and bold.

11. Line 228 and 237, Change to $P < 0.001$, only report exact p value if $p \geq 0.001$.

Answer: We have now changed all P values smaller than 0.001 to " $P < 0.001$ ", to meet the formatting requirement of the journal.

12. Line 231, the interactive dashboard, is this up to date with the latest revisions?

Answer: We thank the editor for the kind reminder and can confirm that the results presented on the interactive dashboard are the latest version.

13. **Fig 2.**, also change to lower case in the figure itself.

Answer: As suggested by the editor, we have now changed all panel labels of **Fig. 2** to lower case and bold. We also went through the supplementary materials and changed the panel labels of all figures to lower cases.

14. Fig. 3, Panel labels missing in the figure. Also, is it possible to increase font sizes by 2 points in this figure?

Answer: We thank the editor for pointing this out and have now added panel labels to **Fig. 3** and increased font sizes as suggested.

15. Discussion, please make sure to speak of associations rather than causes/impacts, as the nature of the evidence does not allow for strong causal inference.

Answer: We thank the editor for pointing this out. We have now gone through the manuscript and revised sentences with similar issues.

16. "We also acknowledge that women with existing diseases, such as women with heart disease, can experience serious complications from pregnancy, thereby noting that strategies for high-risk individuals need further development⁵³." This seems out of place in a paragraph that describes eugenics and discrimination.

Answer: We thank the editor for the comment and have now removed this sentence from the manuscript.

17. "Given that estimates of the voluntarily childfree group are around a quarter of the total population of people without children, we note that public health interventions should take this group of voluntarily childfree individuals into account." In what way?

Answer: We thank the reviewer for the comment and have now rephrased the sentence as follows (**Manuscript p14, Line 435-438**):

"Given that estimates of the voluntarily childfree group are around a quarter of the total population of people without children, we note that public health interventions should take this proportion into account and target those who are involuntarily childless due to certain diseases."

18. Methods, please start this section with a description of the datasets and how they were obtained. Please note that, as linkage of data in different databases was performed for this study, and some of the information is private, we require that ethics approval be sought for the study; or evidence of approval from the database controllers for this

specific linked datasets/project be provided. Please specify this information in your manuscript.

Answer: We have now added the following descriptions to the beginning of the **Methods** section (**Manuscript p15, Line 475-480**):

“This study used nationwide registers from Finland and Sweden. The use of Finnish registry data is approved by Digital and population data service agency (VRK/6551/2019-1 and VRK/6551/2019-2), Statistics Finland (TK-53-1813-19), and Finnish Institute for Health and Welfare (THL/804/5.05.00/2019). The use of Swedish registry data is approved by socialstyrelsen (27035/2018) and Statistics Sweden (247849). Ethics committee/IRB of Regional Ethical Review Board in Uppsala gave ethical approval for this work (2018/223).”

19. Data availability statements are publicly available and machine readable, whereas supplementary information files are not. Please provide all information on how to access the data within the data availability statement rather than in the SI.

Answer: We thank the editor for the suggestion and have now updated the data availability statement as follows (**manuscript p19, Line 608-618**):

“All results (aggregated data) can be explored on the interactive online dashboard available at <https://dsgelrs.shinyapps.io/DiseaseSpecificLRS/>. Due to data protection regulations, we are not allowed to share individual-level data directly. However, all Finnish and Swedish register data used in this study can be applied from national data agencies including Statistics Finland (https://www.stat.fi/index_en.html), Population Information System (DVV, <https://dvv.fi/en/individuals>), Finnish Institute for Health and Welfare (THL, <https://thl.fi/en/web/thlfi-en/statistics-and-data/data-and-services/register-descriptions/care-register-for-health-care>), and Finnish Cancer Registry (<https://cancerregistry.fi/>) from Finland, and Statistics Sweden (<https://www.scb.se/en/>) and National Board of Health and Welfare (Socialstyrelsen, <https://www.socialstyrelsen.se/en/>) from Sweden.”

20. Code availability statement. This needs to include a link to your analysis scripts if you can share them publicly.

Answer: The codes for conducting this study are available at <https://github.com/dsgelab/Lifetime-Reproductive-Success>.

Final Decision Letter:

Dear Dr Ganna,

We are pleased to inform you that your Article "Evidence from Finland and Sweden on the relationship between early-life diseases and lifetime childlessness in men and women", has now been accepted for publication in Nature Human Behaviour.

Please note that *Nature Human Behaviour* is a Transformative Journal (TJ). Authors may publish their research with us through the traditional subscription access route or make their paper immediately open access through payment of an article-processing charge (APC). Authors will not be required to make a final decision about access to their article until it has been accepted. Find out more about Transformative Journals

Authors may need to take specific actions to achieve compliance with funder and institutional open access mandates. If your research is supported by a funder that requires immediate open access (e.g. according to Plan S principles) then you should select the gold OA route, and we will direct you to the compliant route where possible. For authors selecting the subscription publication route, the journal's standard licensing terms will need to be accepted, including self-archiving policies. Those licensing terms will supersede any other terms that the author or any third party may assert apply to any version of the manuscript.

Once your manuscript is typeset and you have completed the appropriate grant of rights, you will receive a link to your electronic proof via email with a request to make any corrections within 48 hours. If, when you receive your proof, you cannot meet this deadline, please inform us at risproduction@springernature.com immediately. Once your paper has been scheduled for online publication, the Nature press office will be in touch to confirm the details.

To assist our authors in disseminating their research to the broader community, our SharedIt initiative provides you with a unique shareable link that will allow anyone (with or without a

subscription) to read the published article. Recipients of the link with a subscription will also be able to download and print the PDF.

With best regards,

Arunas Radzvilavicius, PhD
Senior Editor, Nature Human Behaviour
Nature Research